# High-resolution silkworm pan-genome provides genetic insights into artificial selection and ecological adaptation

The silkworm *Bombyx mori* is an important economic insect for producing silk, the "queen of fabrics". The currently available genomes limit the understanding of its genetic diversity and the discovery of valuable alleles for breeding. Here, we deeply re-sequence 1,078 silkworms and assemble long-read genomes for 545 representatives. We construct a high-resolution pan-genome dataset representing almost the entire genomic content in the silkworm. We find that the silkworm population harbors a high density of genomic variants and identify 7308 new genes, 4260 (22%) core genes, and 3,432,266 non-redundant structure variations (SVs). We reveal hundreds of genes and SVs that may contribute to the artificial selection (domestication and breeding) of silkworm. Further, we focus on four genes responsible, respectively, for two economic (silk yield and silk fineness) and two ecologically adaptive traits (egg diapause and aposematic coloration). Taken together, our population-scale genomic resources will promote functional genomics studies and breeding improvement for silkworm.

Genomic data greatly accelerated the development of biological research in the last two decades. Recently, the focus of genome research has shifted from a single reference genome to a pan-genome approach that provides greater insights into the entire genomic content of a species[1–6]. Taking advantage of the highly efficient detection of structural variants (SVs; >50 bp) and falling costs, third-generation sequencing (TGS) technologies have started to supersede next-generation sequencing (NGS) technologies in the construction of pan-genomes. Recently, increasing long-read based pan-genomes of a single plant or animal species, including soybean[7], tomato[8], rice[9], apple[10], rapeseed[11], *Drosophila*[12,13], butterfly[14], and human being[15,16] have been constructed, facilitating insight into intra-species genomic variation and its contributions to trait determination.

Silkworm, *Bombyx mori*, is an economically important insect that was domesticated from its wild ancestor, *B. mandarina*, ~5000 years ago[17]. The reference silkworm genome (~450 Mb) was first released in 2004 and subsequently updated twice, greatly facilitating functional genomics studies in silkworm and other insects[18–21]. To date, more than one hundred silkworm accessions have been re-sequenced by NGS technology[22]. However, due to the scarcity of wild silkworms and

technical limitations in the previous studies, many trait-associated sites might be missing and furthermore, SVs remain to be explored.

Here, we deeply re-sequence 1078 silkworm strains, generate long-read assemblies for 545 accessions of those strains, and construct a high-resolution pan-genome. We identify hundreds of SVs and genes potentially underlying domestication and breeding of silkworm. We also provide four examples showcasing the utility of these genomes to decode the genetic variation related to key traits. These comprehensive silkworm genome resources will facilitate the basic research and precise breeding in silkworm and enlighten pan-genome studies on other species.

## Results

### Deep resequencing of 1078 silkworms

To explore the full genomic diversity within silkworm (including *B. mori* and *B. mandarina*), we collected as comprehensively as possible 1078 samples of silkworms comprising 205 local strains, 194 improved varieties, and 632 genetic stocks of domestic silkworm (*B. mori*), and 47 wild silkworms (*B. mandarina*) (Fig. 1, Fig. 2a, Supplementary Data 1). A total of 31.52 Tb NGS reads with an average sequencing depth

✉e-mail: xltong@swu.edu.cn; lucheng@swu.edu.cn; zxtian@genetics.ac.cn; wwang@mail.kiz.ac.cn; xbxzh@swu.edu.cn; fydai@swu.edu.cn

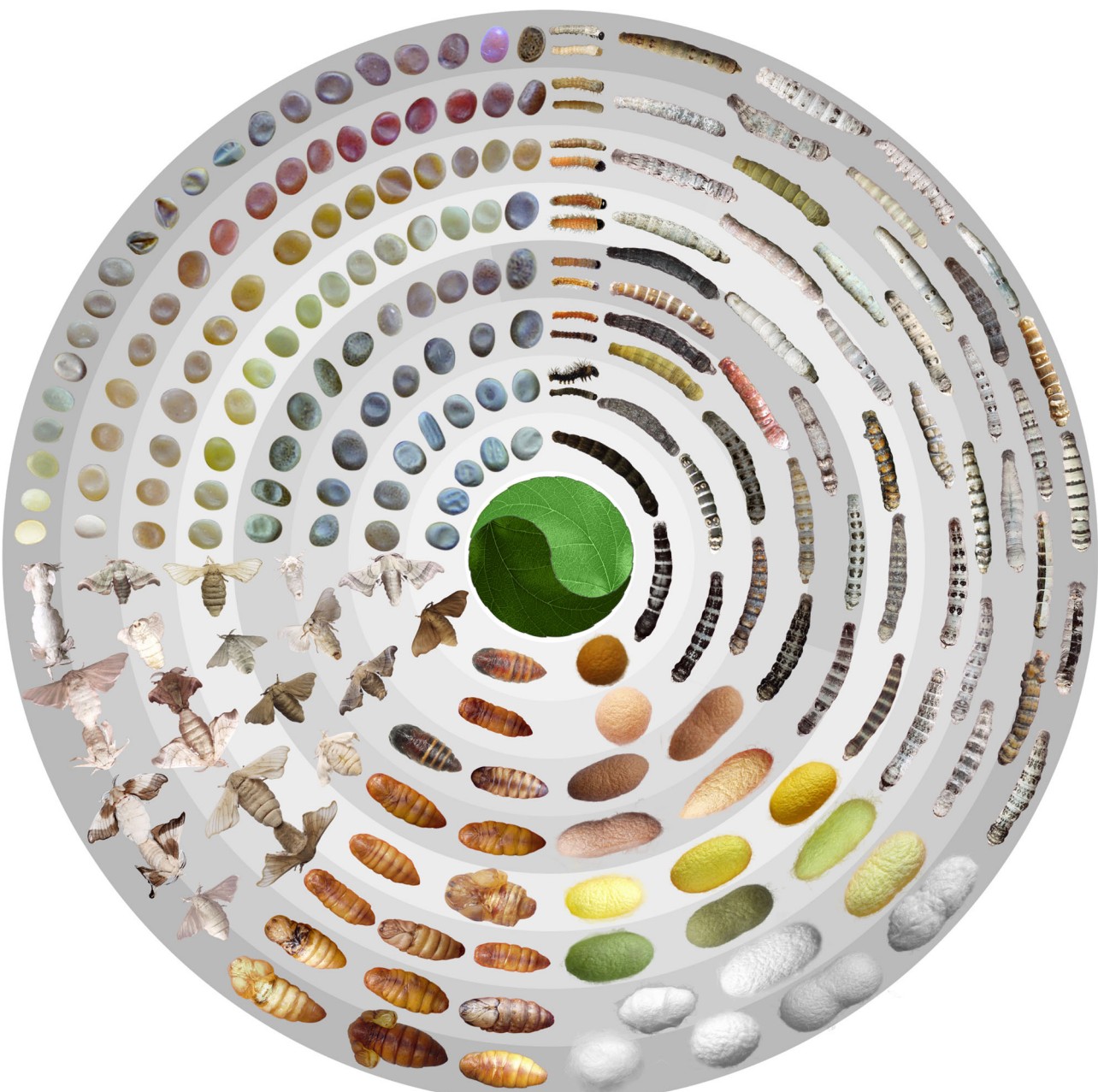

**Fig. 1 | Phenotypic diversity in silkworms.** The enormously diverse phenotypes of silkworms are displayed throughout all developmental stages from the eggs to larvae and pupae (including cocoons and pupae) and adults (clockwise from the north-west quadrant).

of ~65× per sample was obtained (Supplementary Data 1). Using NGS data of the 1078 silkworms plus four previously released wild silkworm genomes[23,24], 43,012,261 single-nucleotide polymorphisms (SNPs) and 9,344,375 small insertions or deletions (Indels, <50 bp) were identified. The SNP and Indel density were one SNP per 11 bp and one Indel per 49 bp, indicating a high degree of genomic diversity in silkworms.

Principal component analyses (PCA) based on whole-genome SNPs of the 1082 genomes showed that PC1 splits individuals into wild and domestic groups, while PC2 further divide individuals into groups based on their geographic origin in general (Supplementary Fig 1a). Furthermore, phylogenetic analysis, like PCA, showed that the 1082 strains are divided into wild group versus domestic population and further subdivided into the subclusters China-local, Europe-local, Tropical-local, and the improved strains in China (CHN-I) and in Japan (JPN-I) (Fig. 2b). Of note, the genetic stock strains are widely distributed within the different subclades of the domestic silkworm clade

(Fig. 2b) and cover therefore broadly across the diversity of domesticated silkworm. Our results show the existence of four major subgroups (wild, China-local, CHN-I, and JPN-I), like the results reported previously[22], but also provide three new insights. First, Europe-local strains are divided into two distinct subclades with some of them close to China-local and others close to the Japan-local resources (absent in the prior study), implying that after the introduction of Chinese silkworm into Europe, silkworm trade also happened between Japan and Europe. Secondly, the improved strains are highly concentrated in two groups, CHN-I and JPN-I, which reflects the narrow genetic base of commercial silkworms (Fig. 2b), while the genetic stock strains present extensive genetic diversity and harbor some special valuable breeding traits, such as disease resistance, excellent feeding performance, or special silk properties. These results suggest that the exploitation and utilization of these abundant genetic resources are essential for future silkworm breeding. Thirdly, the local strains from the middle and lower

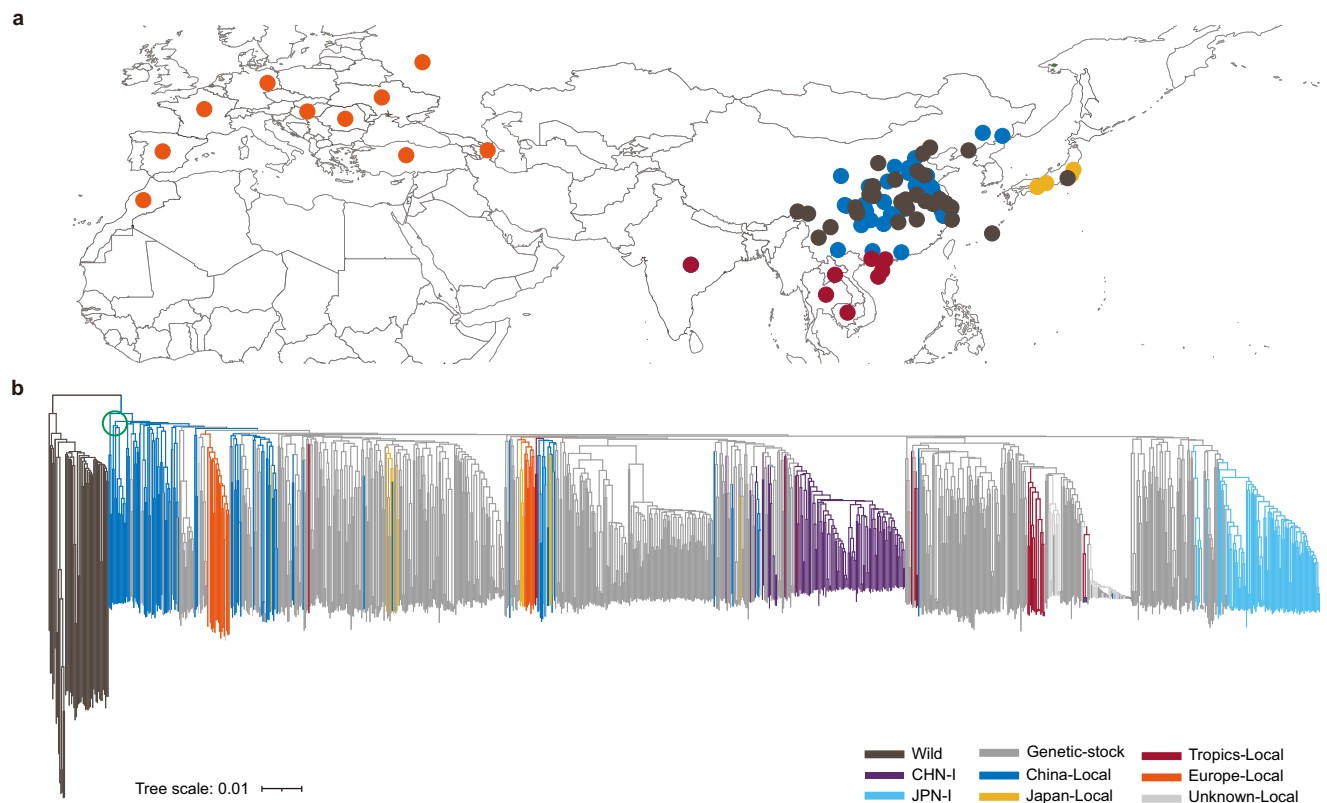

**Fig. 2 | Geographic distribution and phylogenetic tree of silkworm.**
**a** Geographic distribution of 1082 silkworms. **b** Phylogenetic tree based on
genome-wide SNPs of 1082 samples. The clades in the green circle represent the
local strains from the middle and lower reaches of the Yellow River area. The same
color of circles and lines in **a** and **b** represents the same silkworm cluster. CHN-I
(purple), improved strains in China; JPN-I (light blue), improved strains in Japan;
China-Local (blue), local strains in China; Europe-Local (orange), local strains in
Europe; Japan-Local (yellow), local strains in Japan; Tropics-Local (red), local strains
in the tropic region (Guangdong and Guangxi provinces of China, South Asia, and
Southeast Asia); Wild (black), wild silkworm; Genetic stock (gray); Unknown-Local
(light gray), local strains without geographic information.

reaches of the Yellow River located at the basal position of the
domestic silkworm clade (Fig. 2b), which implies that silkworms were
domesticated from this single geographical site, a conclusion further
supported by archeological evidence[25–28].

## Long-read genomes of 545 silkworms

To present an overview of genomic content in silkworms, 545 repre-
sentatives from each group evenly covering the phylogenetic tree were
selected to perform long-read sequencing (nanopore platform) and
genome assembly (Fig. 3a, Supplementary Fig. 1b). We obtained a total
of 24.06 Tb TGS read data with an average sequencing depth of 97×
and an average read length of 23 kb (Fig. 3b, c). De novo assemblies of
these 545 genomes revealed an average genome size of 457.9 Mb, an
average contig N50 size of 7.6 Mb (about half the average length of a
silkworm chromosome), and about two chromosome-level contigs per
genome (Fig. 3d). In addition, repetitive elements constituted 46-49%
of the genomes with an average of 47% (Fig. 3e, Supplementary Data 2).
Among the repetitive elements, non-LTR transposons, including long
interspersed elements (LINEs) and short interspersed elements
(SINEs), were the most abundant, accounting for 23-28% of the gen-
omes with an average of 26%, a value like that in the reference genome.
The BUSCO evaluation value and mapping ratio of NGS reads to the
assembled genomes were 98% and 99% on average, including single-
copy, duplicated as well as fragmented genes (Fig. 3f, Supplementary
Data 2), indicating that the assembled genomes have high
completeness.

To evaluate the full landscape of silkworm genes, we first esti-
mated how many genomes are enough to capture the full set of genes
of silkworms using a gradually superimposed approach (see Method).

We found that the pan-genes obtained from 80, 90, and 100 genomes
were similar with a plateau reached at $n = 80$ (Fig. 3g, Supplementary
Fig. 1c). Thus, the pan-genes contained in the 100 annotated genomes
(14 wild, 41 local, 15 improved, and 30 genetic stock genomes) are
representative for the species. These genomes contain an average of
16,234 genes, 687 transfer RNAs, 123 ribosome RNAs, 13 small nuclear
RNAs, and 6432 microRNAs (Fig. 3h, i, Supplementary Data 3). A total
of 19,411 orthogroups of protein coding genes were identified in the
100 genomes, containing 4260 (22%) core (shared by all 100 samples),
6501 (34%) softcore (shared by >90% samples but not all), 8535 (44%)
dispensable (shared by more than one but ≤90% samples), and 115 (1%)
private (present in only one sample) genes (Fig. 3j, Supplementary
Fig. 1d, Supplementary Data 4). Core genes had the lowest $d_N/d_S$ values,
with 98% of the genes containing an InterPro domain (Supplementary
Fig. 1e, f). They were expressed at higher levels and in more tissues than
dispensable and private genes (Supplementary Fig. 1g, h). GO anno-
tations showed that core genes are enriched in genes of transcription
regulator activity and DNA-binding transcription factor activity com-
pared with the other three groups (Supplementary Fig. 1i, j). We further
found that the orthogroups of silkworm core genes displayed the
widest distribution and the highest sequence identity among 24
insects of 10 orders compared to that of the softcore and dispensable
genes (Supplementary Fig. 1k, l, Supplementary Data 5). These results
suggest that the functions of core genes are more conserved and that
they may play important roles in gene regulation.

Of these 19,411 orthogroups, 7308 (38%) are absent in the gene set
of the prior reference genome[21] and are thus newly identified. Around
83% (5807) of the newly identified genes have GO terms or transcrip-
tional evidence (Supplementary Data 4), and ~99% of the new

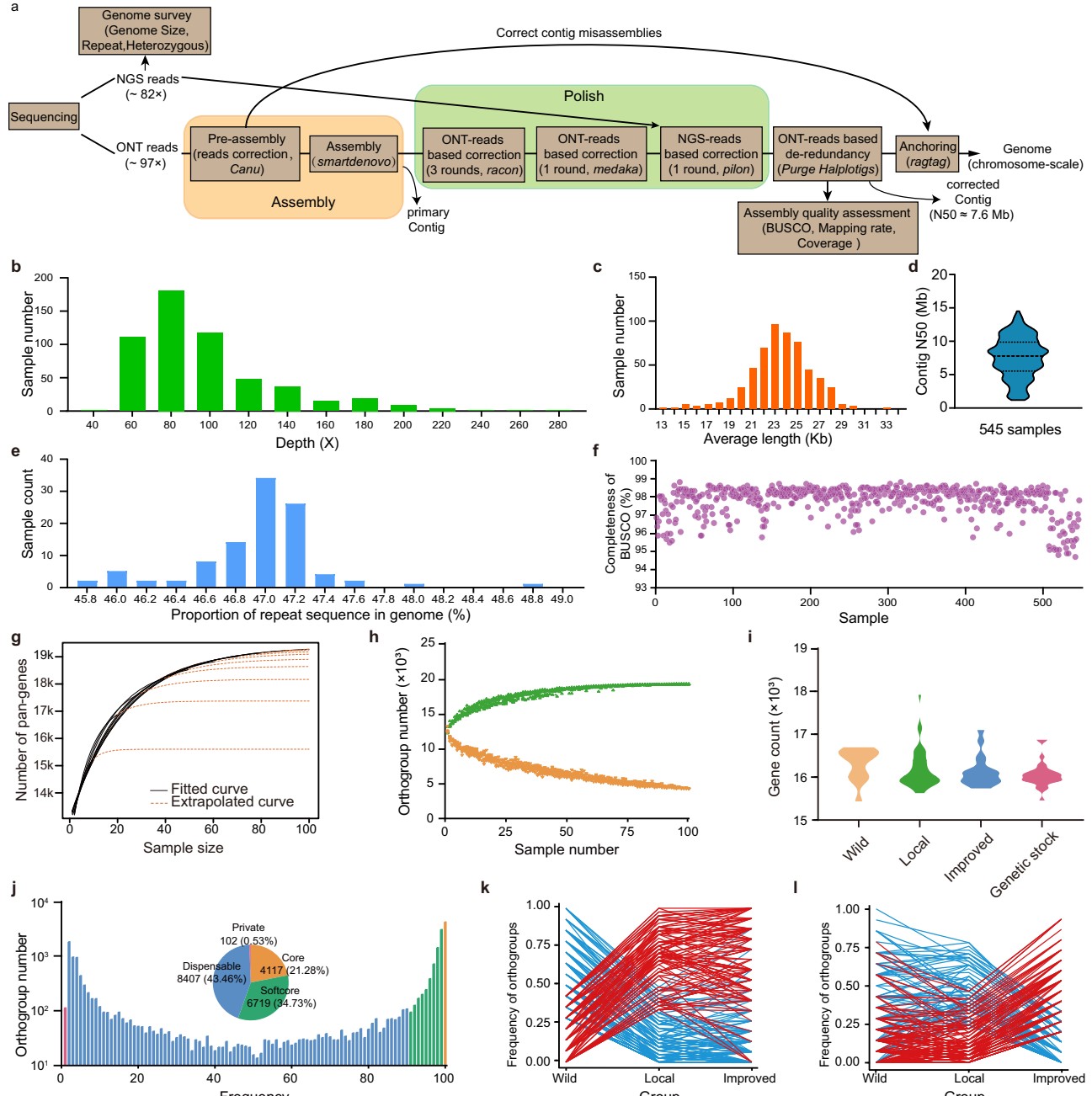

**Fig. 3 | Sequencing, assembly, and pan-gene analysis of 545 silkworm genomes. a** The strategy of genome sequencing and assembling. **b** The average long-read coverage distribution among strains. **c** The average read length distribution among strains. **d** The distribution of contig N50 length among genomes. **e** The proportional distribution of repeat sequences. **f** BUSCO evaluation values. **g** Evaluation of pan-gene plateau. The black curves are fitted with actual data, and the yellow dotted curves are extrapolated by the model of $y = A + Be^{Cx}$. The pan-genes obtained from 80, 90, and 100 genomes are similar. **h** Counts of pan-gene (green) and core-gene (orange) with increased samples. **i** Comparisons of gene counts between wild, local, improved, and genetic stock groups. **j** Frequency of gene number. The pie chart shows the proportion of core, softcore, dispensable, and private genes in those genomes. **k**, **l** genes with significant frequency differentiation between wild-local (**k**) and local-improved comparisons (**l**). In the two comparisons, genes with significantly increased (red) or reduced (blue) frequencies either in domestication (**k**) or improvement (**l**) are shown with different colors.

orthogroups are present in more than two genomes (Supplementary Fig. 1m), suggesting that they are truly present and are informative for further functional genomics studies in silkworm. Distribution analysis showed that 251 and 241 orthogroups had a significant frequency change (FDR < 0.0001 and fold change >2) in comparisons between wild-local as well as local-improved populations (Fig. 3k, l). Among these genes, 72% and 82% were newly identified, which indicates that the newly identified genes have potential roles in silkworm domestication and breeding.

## SV characters and graph-based pan-genome

To construct a high-resolution silkworm pan-genome, we mapped the long reads of each of the 545 genomes to the reference genome[21] and obtained an average of 120,216 SVs per genome (Fig. 4a). The average SV count in wild silkworm genomes is significantly higher ($p < 0.0001$, $t$ test) than in domestic silkworms (Supplementary Data 2). Insertions (INS) and deletions (DEL) (referred to as PAVs, presence/absence variations) constitute ~99% of the SVs (Supplementary Data 2, Fig. 4b). To validate the quality of identified SVs, we checked the previously

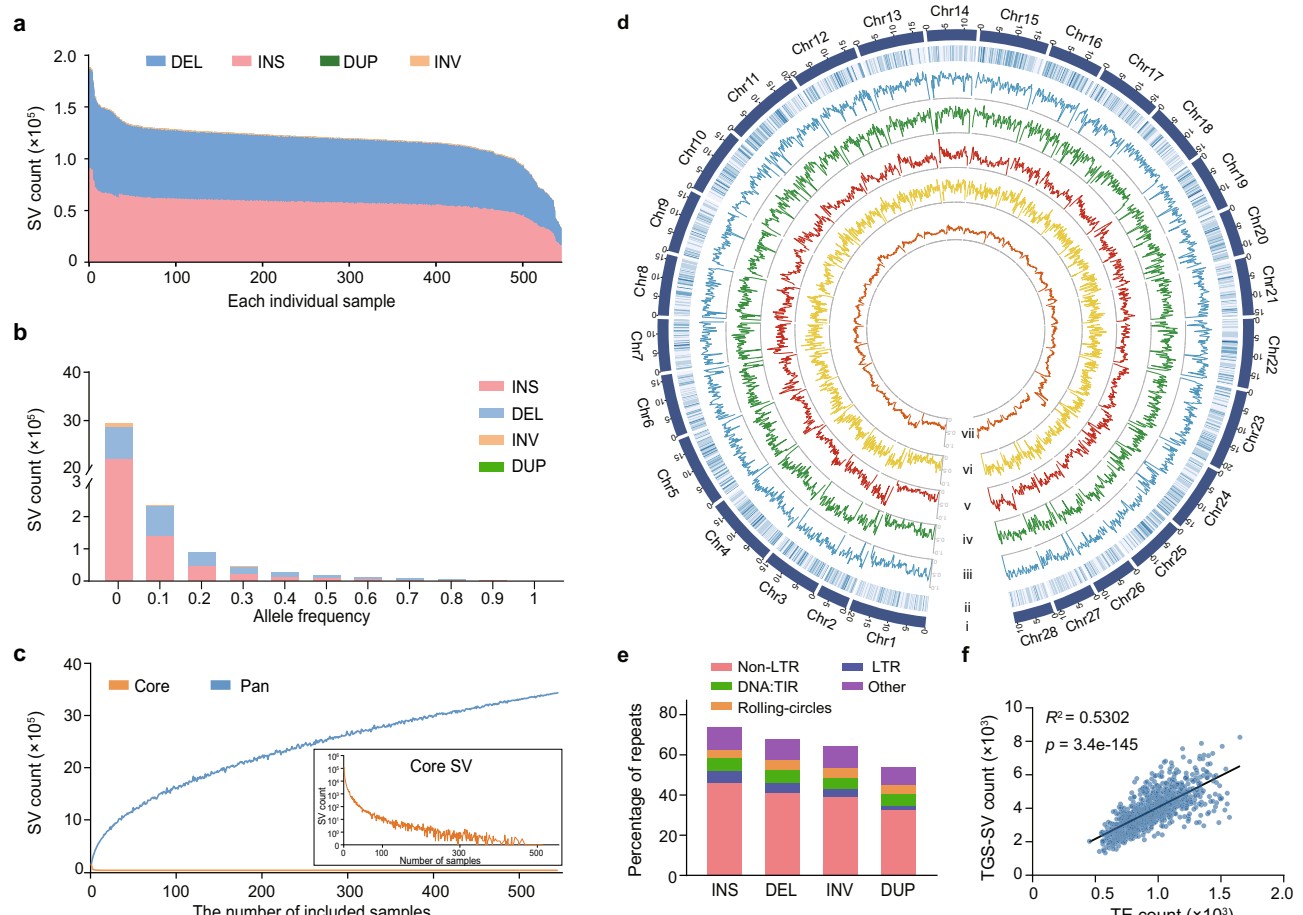

**Fig. 4 | Characterization of SVs in 545 silkworm genomes. a** SV count of insertions (INS), deletions (DEL), duplications (DUP), and inversions (INV) in each of the 545 genomes. The proportions of DUP and INV are too low to be observed in the graph. **b** Allele frequency of nrSVs from 545 samples. **c** Pan-SV and core-SV counts with additional genomes. **d** Distribution map of genetic variations in 1082 genomes. (i) Chromosomes. (ii) Gene density. (iii) SNP and (iv) Indel densities across 1082 genomes. (v) Non-redundant SVs density of 545 TGS genomes. (vi) Density of non-redundant SVs of 537 NGS-only genomes. (vii) TE density. **e** Components of transposable elements (TEs) in sequences of insertion (INS), deletion (DEL), inversion (INV), and duplication (DUP). **f** Correlations of TGS-SV and TE counts. TGS-SV and TE numbers were counted in uninterrupted 500 kb windows. There is a significant linear relationship between the SV and TE distributions on chromosomes (Linear regression, $R^2 = 0.53$, Pearson's $r = 0.7281$, $p = 3.426e-145$, $F$ test). Source data are provided as a Source Data file.

experimentally verified SVs in our sequenced strains and found that all nine reported SVs were present in our identified SVs[29–37] (Supplementary Data 6). Next, we randomly selected 50 SVs to perform polymerase chain reaction (PCR), which confirmed 48 newly identified SVs (96%) (Supplementary Fig. 2, Supplementary Data 7). These results imply that the SV calling was reliable in general. All SVs were merged to generate 3,432,266 non-redundant SVs (nrSVs) with a majority (96%) shorter than 15 kb length and a large proportion (81%) of rare alleles (allele frequencies less than 0.05) (Fig. 4b, Supplementary Fig. 3a, b), presenting an open silkworm pan-SV (Fig. 4c, Supplementary Fig. 3c) and a high SV density (one SV per 134 bp) in the silkworm population (Fig. 4d).

The average SV count per genome in silkworms is, to our knowledge, ~60 and six times higher than in *Drosophila*[12,13,38] and human[15,16]. A primary explanation could be that the silkworm genome harbors a higher density (2075 copies per Mb per genome) of transposable elements (TEs) compared with *Drosophila*[39] (229 copies per Mb per genome) and human[40] (1222 copies per Mb per genome), since TEs are regarded as the major contributor to genomic SVs. Indeed, we found that TEs constituted the largest component (67%) of SV sequences in silkworm genomes (Fig. 4e). Further, the distribution of SVs on chromosomes related significantly to that of TEs ($R^2 = 0.53$, $p < 0.0001$) (Fig. 4f). Another possible explanation could be that SV events occur with a high mutation rate, as reflected by the appearance of a high proportion (47%) of multiple SV alleles (ranging from 2 to 135 nrSVs in a

single site) in all the silkworms (Supplementary Fig. 3d, e). Moreover, domestic silkworms are more tolerant to deleterious or slightly deleterious mutations (rare alleles, allele frequency <0.05) because they are entirely dependent on humans and under weak natural selection pressure. This speculation is further supported by our finding that the proportion of rare allele SVs in domestic silkworm (70%) is higher than in wild silkworm (56%).

We constructed a graph-based pan-genome by integrating all PAVs into the linear reference genome[21]. We mapped the short reads of each long-read sequencing strain to the pan-genome and found that the average values of the precision, recall, and F1 scores were 0.88, 0.74, and 0.80, indicating that the SVs identified on the basis of long reads are reliable. Further, mapping the short reads of the remaining 537 NGS sequenced genomes to the pan-genome led to the identification of 454,671 SVs, containing 59,037 novel SVs (Supplementary Fig. 3f). The distribution pattern of SVs on chromosomes based on NGS data is consistent with that of SVs identified by TGS data (Fig. 4d, Supplementary Fig. 3g). These results suggest that the pan-genome could be used as a comprehensive reference to analyze genomic variations in the short-read sequenced genomes.

## Impact of SVs on gene expression
To investigate the effects of SVs on gene expression, we first analyzed their relative genomic positions. We found that 55% of the SVs are in

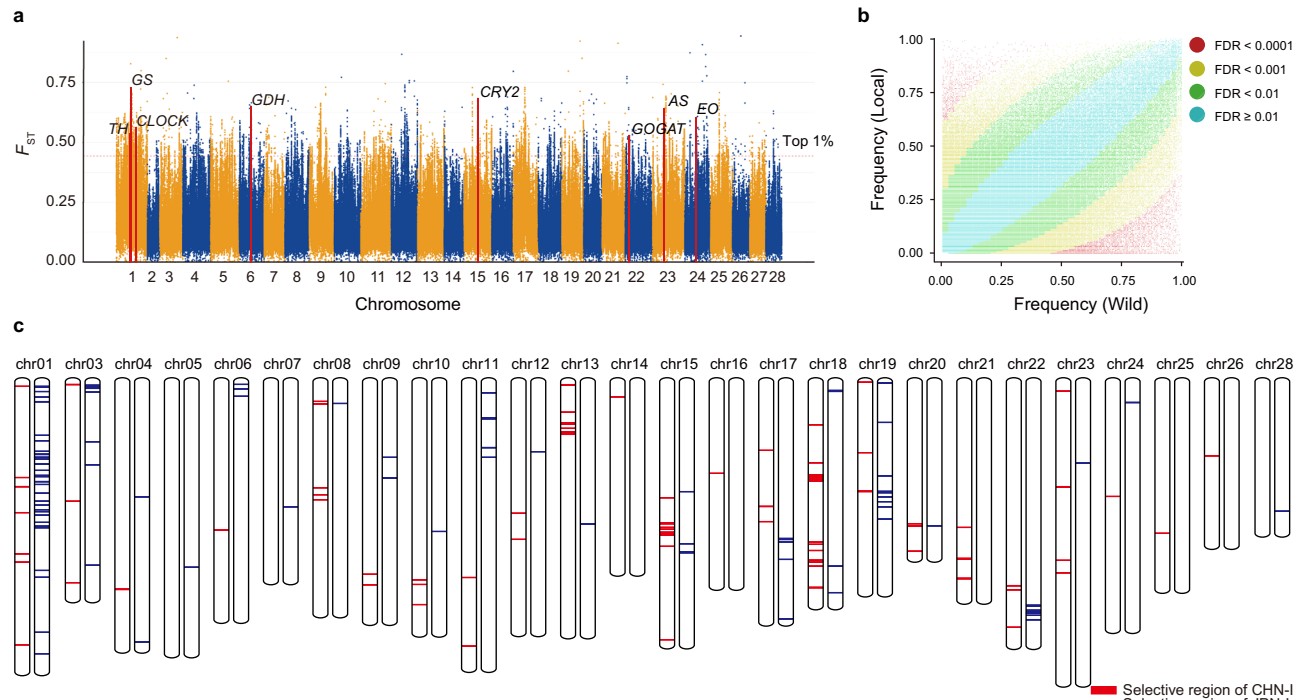

**Fig. 5 | Silkworm domestication and breeding. a** $F_{ST}$ shows selective signal in silkworm domestication. The previously reported domestic genes associated with silk production (*AS*, *GS*, *GDH*, and *GOGAT*), circadian rhythm (*CLOCK*, *CRY2*), development (*EO*), and body color (*TH*) are marked and show obvious selective signal. *TH* (tyrosine hydroxylase), *GS* (glutamine synthetase 2), *CLOCK* (circadian locomoter output cycle protein kaput), *GDH* (glutamate dehydrogenase), *CRY2* (cryptochrome 2), *GOGAT* (glutamate synthase), *AS* (asparagine synthetase), and *EO* (ecdysone oxidase). **b** Frequency distribution of SVs in domestication (in wild and local groups); dots represent SVs. We identified 5,353 SVs (red dots) which have potentially played roles in domestication of silkworm because they show differences in their AFs in wild and domestic silkworms (FDR < 0.0001, fold change >2). **c** Selective regions of CHN-I (red, Chinese improved strains) and JPN-I (blue, Japanese improved strains) resulting from the process of breeding.

potential expression regulatory regions (including the introns and the ±5 kb flanking regions of a gene, as analyzed in this study) or coding sequence (CDS) of reference genes in the entire silkworm population. Among these SVs, we found ~93% (1,762,169) in potential expression regulatory regions (Supplementary Fig. 3h) with the other 7% (130,669) influencing the CDS of 12,661 genes (75% of reference genes). We next investigated gene expression using RNA-seq data of 84 samples from fourteen strains that harbored 178,309 SVs in potential expression regulatory regions, forming 26,188 SV-gene pairs (for each pair, at least three strains with and three strains without the SV). Among these pairs, 2396 SV-gene pairs (9.2%) contain a total of 1560 genes that show differential expression (FDR < 0.001) in at least one tissue between strains with and without corresponding SVs (Supplementary Fig. 3h).

**Genes and SVs underlying silkworm domestication and breeding**
The identification of the artificially selected genes and SVs that underlie silkworm domestication and breeding will facilitate the understanding and improvement of desirable traits in silkworms. We calculated the population divergence index ($F_{ST}$), neutrality tests (Tajima's D), and a cross-population composite likelihood ratio test (XP-CLR) using SNP markers. We defined the intersections of $F_{ST}$, Tajima's D and XP-CLR as selective sweep regions and identified 468 (2.8% of the whole-genome genes) domestication-associated genes (Fig. 5a, Supplementary Data 8a), containing 264 newly identified domestication-associated genes compared with previous studies[22,41,42]. These 468 genes are enriched in amino-acid metabolism, nitrogen metabolism, and circadian rhythm, probably caused by human selection on silkworm growth and development, synthesis of silk protein, and environmental adaptation (Supplementary Fig. 4a). We further compared the allele frequencies of SVs between wild and local silkworms, and identified 5353 domestication-associated SVs (presenting

significant divergent frequencies between wild and local groups) (Fig. 5b). A total of 872 domestication-associated SVs were found in or close to domestication-associated genes, predominantly (95%) distributed in their potential expression regulatory regions (Supplementary Data 8a).

We next performed comparative analyses among improved and local strains to identify breeding associated genes. CHN-I and JPN-I are the improved groups currently used for generating heterosis (or hybrid vigor). We identified 126 (CHN-I) and 116 (JPN-I) improvement-associated regions containing 106 and 92 improvement-associated genes (Fig. 5c, Supplementary Data 8b). Compared with a previous study[22], 185 improvement-associated genes were newly identified. Interestingly, the two improved groups shared only around 3% of these improvement-associated regions (Fig. 5c), suggesting that breeding proceeded independently in CHN-I and JPN-I. These results reveal parts of the genetic bases of silkworm heterosis and provide potential targets for improvement in silkworm breeding. To identify improvement-associated SVs (specifically those showing significant divergent frequencies between improved and local groups), we compared allele frequencies of SVs between local and improved silkworms. We detected 3574 and 3516 improvement-associated SVs in local versus CHN-I and local versus JPN-I comparisons (Supplementary Fig. 4b). In the genomic and flanking (±5 Kb) regions of improvement-associated genes, we identified 312 improvement-associated SVs, most of which (99.7%) are within the potential expression regulatory regions of these genes (Supplementary Data 8b).

**Identification of loci related to commercially important traits**
Artificial selection of silkworm has focused on the economically important silk-related traits, such as yield or quality of silk. But few causal genes and loci for these commercially valuable traits are

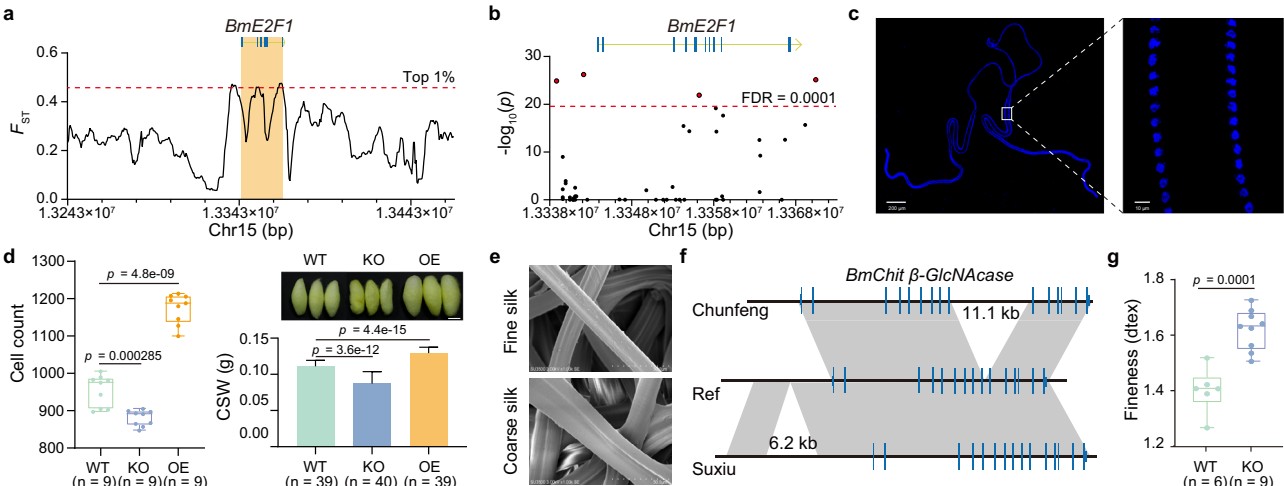

**Fig. 6 | Genetic basis of silkworm economic traits. a, b** The genomic region of *BmE2F1* shows a signature of a positive selection (**a**) with four SVs (red dots) showing a significant frequency divergence (**b**) between local and CHN-I silkworms. **c** Embryonic silk gland nuclei stained with DAPI (blue fluorescence). **d** Silk gland cell count of *BmE2F1* knockout (KO) line, *BmE2F1* overexpression (OE) line, and wild type (WT, Dazao) lines (left histogram). Cocoon shell weight (CSW) and cocoons of *BmE2F1* KO, *BmE2F1* OE, and WT lines (right histogram and picture of cocoons). Data are shown as mean ± SD. Scale bar, 1 cm. Student's *t* test (two-tailed). **e** Silk

with fine and coarse denier under a scanning electron microscope. **f** An 11.1 kb insertion in intron and a 6.2 kb downstream insertion of the *BmChit β-GlcNAcase* gene in the fineness strains Chunfeng and Suxiu. **g** CRISPR-cas9 mediated knockout of *BmChit β-GlcNAcase* produced coarser silk. Student's *t* test (two-tailed). In box plots, horizontal lines within boxes indicate the medians, box boundaries indicate the 1st and 3rd quartiles, and whiskers indicate the minima and maxima. Source data are provided as a Source Data file.

characterized so far. Taking advantage of the high-resolution pan-genome, we explored the genes and variations contributing to those desirable breeding traits.

Silk yield is largely affected by the number and endoreplication of silk gland cells synthesizing the silk proteins. Among the improvement-associated genes, we found that the transcription factor *BmE2F1* involved in cell cycle progression shows a significant selection signal during breeding (Fig. 6a, Supplementary Fig. 5a). The *BmE2F1* gene harbors four improvement-associated SVs including one deletion and three insertions in its *cis*-regulatory region and introns (Fig. 6b, Supplementary Fig. 5b). The allele frequencies of these four SVs are significantly higher in improved silkworms (FDR < 0.0001, fold change >2) than in local silkworms (Supplementary Fig. 5b). The silk yield of the strains harboring the four improvement-associated SVs is significantly higher than that of the strains without these SVs (Supplementary Fig 5c). Further, CRISPR-cas9 mediated knockout of *BmE2F1* reduces the number of silk gland cells by 7.68% and silk yield by 22%. In contrast, the transgenic overexpression of *BmE2F1* increases the number of silk gland cells by 23% and silk yield by 16% (Fig. 6c, d, Supplementary Fig. 5d, e). These results indicate that the *BmE2F1* gene participates in the determination of the number of silk gland cells, thereby affecting silk yield.

Fine silk has higher economic value in sericulture, but the genetic basis of fiber fineness is unknown. We previously found that a part of the spinneret, the silk press, is related to fineness[43]. Here we performed RNA-seq of the silk press in four strains including two fine silk strains (Suxiu, Chunfeng) and two coarse silk strains (Xiafang, Qiubai) (Fig. 6e) and identified 40 differentially expressed genes (DEGs) (Supplementary Fig. 6a). We scanned the variations in the genomic regions of these DEGs to identify SVs that are unique to the fine silk strains relative to the coarse silk strains and rare in the entire silkworm population. We found an 11.1 kb intron insertion and a 6.2 kb downstream insertion of the *chitooligosaccharidolytic beta-N-acetylglucosaminidase (BmChit β-GlcNAcase)* gene in Chunfeng and Suxiu strains (Fig. 6f). We found that the *BmChit β-GlcNAcase* gene is expressed at a significantly higher level in fine silk strains (Suxiu, Chunfeng) and has an expression peak in silk press at the wandering stage (which occurs at the start of spinning) (Supplementary Fig. 6b, c). CRISPR-cas9 mediated knockout of the

*BmChit β-GlcNAcase* gene produced coarser silk (Fig. 6g, Supplementary Fig. 6d). All these results suggest a key role of the *BmChit β-GlcNAcase* gene in the determination of silk fineness.

## Decoding insect adaptation-related genes using the pan-genome

Diapause is a common adaptive strategy that ensures the survival of organisms under deleterious environmental conditions[44]. Although the diapause hormone (DH), an embryonic diapause inducer, was first discovered in silkworm[45], little information is available concerning the embryonic diapause genes.

The *pnd* strain produces non-diapause homozygous (*pnd/pnd*) eggs and diapause heterozygous (*pnd/+*) offspring that are determined by a genetic factor in chromosome 11 (11–55.89 cM) after fertilization, but not by the diapause factor during oogenesis[46–49]. To identify the embryonic diapause gene, we searched the genomic variations in the *pnd/pnd* homozygote (BomM479) within the region between the previously reported *black pupa (bp)* locus (11-42.5 cM, KWMTBOMO06855)[34] and the end of chromosome 11. We found 10 genes with exonic sequence variation in the investigated range (Supplementary Fig. 7a). According to the gene function annotation, a gene (KWMTBOMO06872, *BmTret1-like*) encoding a sugar transporter appeared to be an ideal candidate because trehalose transport is crucial for insect diapause[50]. A 747 bp deletion was found in the 3′-untranslated region (3′-UTR) of *BmTret1-like* in *pnd* homozygotes, while both the mutant and normal copies are detectable in heterozygotes (*pnd/+*) (Fig. 7a; Supplementary Fig. 7b). During the early embryonic stage, we observed that the expression level of *BmTret1-like* in homozygotes (*pnd/pnd*) is significantly lower (*p* < 0.01, *t* test) than that in heterozygotes (*pnd/+*) (Fig. 7b). To test the function of *BmTret1-like*, CRISPR/Cas9 mediated knockout was performed in bivoltine strain Dazao (a strain that generates diapause eggs when the maternal embryos are incubated at 25 °C). The injected eggs and their offspring were always incubated at 25 °C. After three generations of hybridization and mutation screening, we obtained three batches of *BmTret1-like⁻/⁻* homozygotes that resulted in a non-diapause phenotype (Fig. 7c). These results suggest that *BmTret1-like* is a critical embryonic diapause determinant after fertilization.

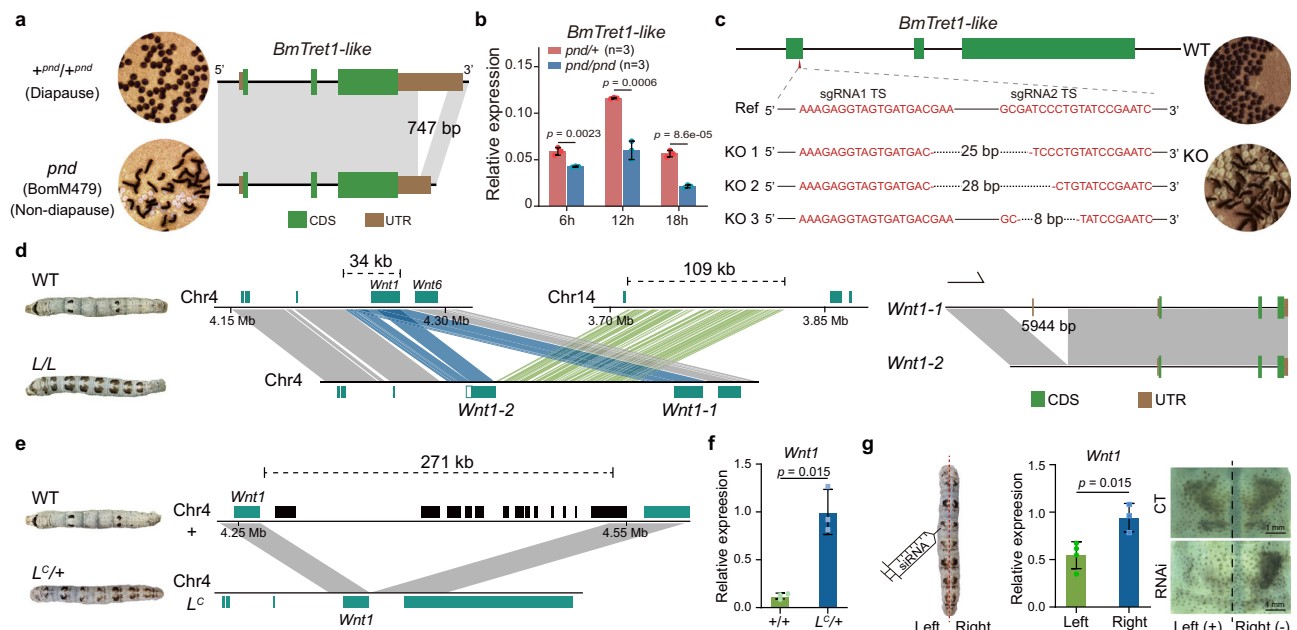

**Fig. 7 | Genetic basis of adaptive traits. a** When compared to diapause wild type ($+^{pnd}/+^{pnd}$), a 747 bp deletion can be identified in the 3'-UTR of *BmTret1-like* in non-diapause *pnd*. **b** In embryonic stages at 6, 12, and 18 hours, the expression level of *BmTret1-like* in homozygotes (*pnd/pnd*) is significantly lower than that in heterozygotes ($+^{pnd}/pnd$). Three biological duplications were conducted. Data are shown as mean ± SD. **c** CRISPR/Cas9 mediated *BmTret1-like* knockout (KO) in wild type (WT, Dazao) individuals. Three *BmTret1-like*$^{-/-}$ lines, KO1, KO2, and KO3 contain a 25 bp, 28 bp, and 8 bp deletion in *BmTret1-like*. The *BmTret1-like* knockout lines generate non-diapause eggs under 25 °C. The wild-type bivoltine strain (WT, Dazao) generates diapause eggs when the maternal embryos were incubated under 25 °C. **d** Large insertion and duplication in the *L* locus and the difference between *Wnt1-1* and *Wnt1-2*. **e** Large deletion in $L^C$ locus. **f** Expression of *Wnt1* in +/+ and $L^C$/+, three biological duplications were conducted. Data are shown as mean ± SD. **g** *Wnt1* expression and phenotype after *Wnt1* RNAi (*n* = 20). CT, control ($L^C$/+). Scale bar, 5 mm. Student's *t* test (two-tailed). Data are shown as mean ± SD. Source data are provided as a Source Data file.

Aposematic coloration, the presence of conspicuous body markings acting as signals, is another important adaptive trait. Two silkworm allelic mutants, *Multilunar* (*L*) and *Caltrop-type multilunar* ($L^C$) (Fig. 7d, e), lead to similar twin-spot markings commonly used as aposematic markings to avoid predators in caterpillars[51,52]. A prior study speculated that the *L* phenotype could be caused by short sequence changes in the 5'-flanking 19 kb region of *Wnt1* on chromosome 4[53]. In our assembled 545 genomes, nine strains contain an *L* allele (Supplementary Fig. 7c, Supplementary Data 1). Comparative genomic analysis between samples with- and without- *L* allele revealed that two SVs, a duplication (34 kb) containing an additional *Wnt1* copy (named *Wnt1-2*) with a 5944 bp deletion in its *cis*-regulatory region and an insertion (109 kb) derived from chromosome 14 in the 5'-terminal 18 kb site of *Wnt1-1* (original *Wnt1*) (Fig. 7d), are present in each of the nine strains containing an *L* allele but are absent in any strain without an *L* allele. We next investigated SNPs and Indels in the ±20 kb flanking region of the *Wnt1-1* gene and found that only three variants (one Indel and two SNPs) located in the second intron of *Wnt1-1* are specifically present but not fixed in the *L* strains (Supplementary Fig. 7d). These results suggest that the *L* phenotype could be related to these structural variations in the upstream region of *Wnt1-1*.

A prior study revealed that three SNP markers distinguish *Wnt1* transcript in *L* (g01) from non-*L* (N4, p50) strains[53]. Fortunately, two of those SNPs are also able to differentiate *Wnt1-1* and *Wnt1-2* in a single *L* strain, BomM527 (Supplementary Fig. 7e). To investigate which copy of *Wnt1* is expressed in the spot marking region of epidermis in the BomM527 strain, we performed RT-PCR and Sanger sequencing, and detected only the *Wnt1-1* transcript in the epidermis (Supplementary Fig. 7f). Taking these results together, we can speculate that the *L*-specific SVs upstream of *Wnt1-1* may cause its ectopic expression in the spot marking region of the epidermis, resulting in the *L* phenotype.

For the $L^C$ mutant, we found a specific large deletion (271 kb) in the 3'-flanking region of *Wnt1* (Fig. 7e). We observed that the *Wnt1* expression was significantly higher in the epidermis of heterozygous $L^C$ ($L^C$/+) mutants than in normal strains (+/+) (Fig. 7f). Further, *Wnt1* RNAi applied in the left side of the epidermis of $L^C$ ($L^C$/+) larvae blocked spot marking formation (Fig. 7g). These results reveal that large and complex SVs in *L* alleles, which cannot be obtained by map-based cloning, affect the expression pattern of *Wnt1* and result in twin-spot markings.

## Discussion

In this work, we provide a large-scale digital gene bank of silkworm bioresources and find a high level of genetic diversity in silkworms. We have a larger sample size and a wider geographic distribution of sample set than in the previous publications that contain 40 strains (11 wild silkworms and 29 domestic silkworms) in 2009[41] and 144 strains (seven wild silkworms and 137 domestic silkworms) in 2018[22]. We release 1078 high-depth NGS and 545 high-quality reference genomes and construct almost the entire pan-genome of silkworm with 7038 newly identified genes, more than fifty million short variants (SNPs and Indels), and over three million SVs. This dataset enables the functional investigation of many types of sequences. These include many coding genes absent from the existing silkworm reference genome, the core/private genes in silkworms and the variable genes among populations, the complex SVs that are difficult to detect from NGS data, and the genomic variants in the non-coding regions that were rarely touched in previous studies using forward and reverse genetic approaches. These new resources will greatly support high-throughput screening of novel traits for functional genomic research and breeding improvement of silkworms and serve as a guideline for pan-genome study in other species.

Although the domestic silkworm (*B. mori*) is a completely domesticated economic insect entirely dependent on humans for survival, only a few economically important genes have been clearly identified so far. In this study, we identify 468 domestication-associated genes and 189 improvement-associated genes. Compared

with prior studies[22,41,42], the current report represents an additional set of 264 genes associated with domestication and 185 genes associated with improvement. The analysis of the functions of these genes will reveal the genetic basis of artificial selection, provide improvement targets, and promote our understanding of the behavior differences between *B. mori* and *B. mandarina*, such as larval ability to withstand crowding and handling, relatively docile feeding (without a strong drive for finding food), and loss of flight ability. In addition, this study reveals differences between the artificial selection loci of the Chinese and Japanese improvement lines, which will help unravel the underlying basis of hybrid vigor generated from these two populations and aid in the design of better combinations of beneficial loci for subsequent improvement of silkworm. Furthermore, our application of a large-scale pan-genome to decipher two genes that control important economic traits in silkworms may also be used to reveal genetic mechanisms and traits associated with the survival of wild populations and evolution of new species under strong natural selection by human and non-human factors.

Silkworm harbors extensive phenotypic diversity in embryonic and larval phases (Fig. 1); this differs from the well-known model system *Drosophila* that exhibits phenotypic diversity mainly in the adult stage. This feature makes silkworm valuable for studies of morphological diversity in insects. Notably, most of our sequenced silkworms are well described phenotypically, especially for the group that comprises the genetic stocks (Supplementary Data 1). Benefiting from long-term research on silkworm genetics, many loci related to phenotypes have been mapped to the 28 chromosomes of silkworm, generating a comprehensive genetic linkage map[49]. Our pan-genome data, combined with the genetic linkage map, will facilitate the genetic interpretation of intriguing traits and contribute to insect biology. For instance, we found that the ectopic expression of the *Wnt1* (*Wnt1-1*) gene, which is probably caused by structural variations, is responsible for the new color patterning of twin-spot markings on caterpillars. A similar but altered patterning ($L^c$) could also be achieved by the variation of *cis*-regulatory sequences influencing the same gene, *Wnt1*. This finding has increased our understanding of genetic mechanisms underlying the evolution and diversity of color patterns. Thus, the deciphering of the genotype-phenotype relationships in these abundant silkworm resources can further promote our understanding of the genetic architecture of diversification and adaptive evolution in this lepidopteran and may also apply to other insects.

Our results show that the generated pan-genome, based on hundreds of long-read sequenced genomes, provides resources for high-throughput and accurate assessment of valuable alleles for silkworm functional genomics research and breeding. This ushers a new era for silkworm basic research and molecular breeding and provides guidelines for the large-scale pan-genome research of other species.

## Methods

No statistical methods were used to predetermine sample size. The experiments were not randomized, and investigators were not blinded to allocation during experiments and outcome assessments.

### Silkworm collection

We collected 1078 silkworms, representing 1031 domestic silkworms (*B. mori*, including 205 local strains, 194 improved varieties, and 632 genetic stocks) and 47 wild silkworms (*B. mandarina*). For collection of wild silkworms, permits were not required. Local strains are breeding resources that were long maintained, without further selective breeding, in diverse geographic regions of the traditional silk producing countries (e.g., China, Japan, Korea, India, Thailand, Laos, Vietnam, Russia, France, Italy, Germany, Hungary, Spain, Turkey, Romania, Morocco, Cambodia, Azerbaijan, Ukraine, and Bulgaria). Improved varieties showing desirable properties for commercial breeding (e.g.,

high yield and quality silk, natural color cocoon, greater robustness, uniform development, and high hatchability) are strains highly selectively bred for modern sericulture. Genetic stocks are the natural mutants discovered during domestication and breeding improvement process, together with the artificial mutants induced by chemical or physical treatment or genetic engineering. Wild silkworms collected from a full range of their geographic distributions in China represent the ancestor (*B. mandarina*) of the domesticated silkworm. It is worth mentioning that the silkworm genetic stocks maintained since the 1900s constitute a unique resource for silkworm breeding and insect biology. They harbor some special valuable breeding traits, such as adult wings with few scales, excellent feeding performance, high stress tolerance, disease resistance, or special silk properties. More than 400 mutants have been mapped to all 28 linkage groups of silkworms using classical linkage analysis[49].

Most collections (~90%) are from the silkworm genetic resource banks at Southwest University (Chongqing, China). A minority of germplasms was collected from other universities and sericulture research institutes mentioned in the Acknowledgement section.

### Genomic DNA extraction

Pupae or larvae (without midgut) were collected and stored at −80 °C for genomic DNA extraction (Supplementary Data 1). For ONT sequencing, a blood & cell culture DNA midi kit (catalog# 13343, QIAGEN) was used to extract genomic DNA following the manufacturer's procedure. For short-read sequencing, the phenol/chloroform method was used to extract genomic DNA. The extracted DNA pellet was dissolved in 30–200 μL Tris-EDTA (TE) buffer solution for further study.

### Short-read sequencing

For NGS, a paired-end sequencing library of each sample of 1078 silkworms was constructed with insertion sizes ranging from 300 to 400 bp and sequenced through the DNBSEQ platform of BGI (China). *SOAPnuke*[54] v1.5.6 was used to filter out the low-quality reads (a read containing over 40% low-quality bases, base quality value less than 20 was considered as low-quality) and remove PCR duplication reads with parameters -n 0.03 -l 20 -q 0.4 -G -d -Q 2. The two prior resequencing projects of silkworms[22,41] generated low coverage depths (only ~3× and ~13× coverage depth), limiting the utilization of those data in genome assembly and detection of structural variation. Thus, we sequenced all our samples in high depth ranging from 22× to 181× (approximately 65× on average). To ensure a better genome assembly for samples used in long-read sequencing, higher coverage depths ranging from 52× to 181× (82× on average) were generated. The sequencing information for each sample is listed in Supplementary Data 1.

### Long-read sequencing

For long-read DNA sequencing, 545 samples containing 39 wild, 162 local, 117 improved, and 227 genetic stock silkworms were selected to build Oxford Nanopore Technology (ONT) sequencing libraries in BGI (China). According to the standard procedure of library construction of the Oxford Nanopore Technologies Company, 20 kb DNA library of each sample was used to sequence on the PromethION platform with R9.4 chemistry using the Ligation Sequencing Kit SQK-LSK109 (Oxford Nanopore Technologies). For the pooling library of two samples, reads of each sample must be split before filtering. Therefore, *guppy_barcoder* v3.1.50 (https://nanoporetech.com/) and *qcat* v1.0.1 (https://github.com/nanoporetech/qcat) were applied to split reads and the intersection of the two results was used as split data. The *porechop* v0.2.4 (https://github.com/rrwick/Porechop) program was used to find and remove adapters. We also removed the reads with length <5 kb and average quality <7. The final sequencing depth ranged from 48× to 277×. The N50 read sizes ranged from 13.5 to 44.9 kb with an average of 30 kb (Supplementary Data 2).

## Transcriptome sequencing

For RNA-Seq, six tissues including cuticle, fat body, head, hemolymph, midgut, and silk gland of fourteen samples (BomL85, BomL194, BomL41, BomL84, BomP79, BomL114, BomL170, BomL122, BomL31, BomL210, BomL13, BomL112, BomP128, and BomW44) were collected on the 3$^{rd}$ day of the final larval stage. Total RNA was isolated using Trizol (Invitrogen). An RNA-seq library was constructed using the MGIEasy RNA Library Prep Kit. All RNA sequencing was carried out at Frasergen (Wuhan, China) and Novogene (Tianjin, China). The mRNAs of each tissue were collected from more than three individuals, with two biological replicates.

## SNP and Indel calling

The short reads of the 1082 silkworm strains were mapped to the silkworm reference genome by *BWA*[55] v0.7.17 *mem* with default parameters. The *SAMtools*[56] v1.11 and *Picard* v2.23.5 (https://broadinstitute.github.io/picard/) programs were used to filter the unmapped and duplicated reads. A GVCF file of each sample was obtained using the *GATK*[57] v4.1.8.1 *HaplotypeCaller* with the parameter -ERC = GVCF. The GVCF files of all samples were merged with *GATK CombineGVCFs*. The joint-call step of all 1082 samples was performed using *GATK GenotypeGVCFs* to generate a combined VCF file. Finally, the variants were filtered by *GATK VariantFiltration* with the following parameters: -filter "QUAL < 50.0"–filter-name LowQ -filter "DP < 200"–filter-name LowD -filter "DP > 100000"–filter-name HigD–filter-expression "MQ < 40.0, QD < 2.0, FS > 60.0, SOR > 5.0, MQRankSum < −12.5, ReadPosRankSum < −8.0"–filter-name LowQualFilter–missing-values-evaluate-as-failing. A total of 43,012,261 SNPs and 9,344,375 Indels were identified. A previous study identified ~37 million SNPs (the number of Indels was not given) in 144 silkworms[22]. The SNP coordinates were not comparable because we used a different version of the reference genome. However, at least six million SNPs were newly identified in our study.

## Phylogenetic and PCA analysis

*VCF2Dis* v1.42 (https://github.com/BGI-shenzhen/VCF2Dis) was used to estimate *p* distances between every two samples based on a VCF file containing SNPs. *PHYLIPNEW* v3.69 *fneighbor* (http://emboss.sourceforge.net/apps/cvs/embassy/phylipnew/) was used to build the Neighbor-Joining Tree. For PCA analysis, we used *plink*[58] v2.0 (–make-bed) to convert a VCF file to a bed file. Subsequently, the *GCTA*[59] v1.93.0 beta program was used to estimate the genetic relationship matrix (GRM) and calculate eigenvectors.

## Selective sweeps

Artificial selection regions were estimated using a sliding window approach with 5 kb window and 500 bp step size. For each window, we calculated the population divergence index ($F_{ST}$), neutrality tests (Tajima's D), and XP-CLR based on the prior formulas[60–62]. Domestication-associated regions were identified by comparing wild with local silkworms and improvement-associated regions were defined by comparing local with improved silkworms in China (CHN-I) and Japan (JPN-I) separately. The overlapped genome regions of the top 1% $F_{ST}$, top 5% XP-CLR and lowest 5% Tajima's $D_{descendant}$ (and Tajima's $D_{descendant}$ < Tajima's $D_{ancestral}$) signatures were defined as candidate selective sweep regions. The genes included in these regions were regarded as potentially associated with silkworm domestication and breeding.

## Genome assembly

Before assembly, *Jellyfish*[63] v2.2.6 was used to count *k*-mer frequencies (*k* = 17). Then, the genome characters including genome size, repetitiveness, and the rate of heterozygosity were predicted with *genomeScope*[64] v1.0 using NGS data.

The de novo genome assembly was performed with the following pipeline: (a) The ONT reads were corrected by *Canu*[65] v1.8. (b) The corrected ONT reads were assembled into contigs by *Smartdenovo*[66] v1.0 with the following parameters: wtpre −J 5000; wtzmo −k 16 −z 10 −Z 16 −U −1 −m 0.6 −A 1000; wtclp −d 3 −k 300 −m 0.6; wtlay −w 300 −s 200 −m 0.6 −r 0.95 −c 1; wtcns −m 0.6. (c) The contigs were corrected three times with ONT reads by *racon*[67] v1.3.3, and one time by *medaka* v0.7.1 (https://github.com/nanoporetech/medaka). In this step, *minimap2*[68] v2.17 was used to map ONT reads to the contigs, and *racon* was used to correct contigs and generated consensus sequences. The final correction was conducted using the *medaka* program. (d) *pilon*[69] v1.23 software was used to polish the corrected contigs. In this step, NGS reads were mapped to corrected contigs using *BWA mem*, and *pilon* was used to polish with the following parameters:–fix bases–mindepth 20–verbose–diploid. We only used the reads with a mapping quality above 20 for this step. (e) If the assembled genome exceeded the estimated size of genome survey by >5%, we performed de-redundancy using *Purge Haplotigs*[70] v1.0.0. (f) To obtain chromosome-level genomes, we used *RagTag*[71] v1.0.0 to correct and anchor the contigs to chromosomes of the reference genome with the corrected ONT reads. Finally, we mapped NGS reads to the assembled genome, and assessed the genome coverage and mapping ratio (using all mapped reads). We also used Benchmarking Universal Single-copy Orthologues (*BUSCO* v 5.2.1) to estimate the integrity of the assembled genome (including complete single-copy, duplicated, and fragmented BUSCOs) using insecta_odb10[72].

## Repetitive element annotation

For the annotation of repetitive elements, simple sequence repeats were identified by *GMATA*[73] v2.2. Tandem repeats (TRs) were identified by *Tandem Repeats Finder* (TRF)[74] v4.09. Then, we used ab initio, structure, and homology-based methods to annotate TEs. Briefly, *MITE-Hunter*[75] v2 (https://github.com/Adamtaranto/MITE_Hunter_2) was used to search miniature inverted-repeat transposable elements (MITEs). *LTR_finder*[76] v1.07, *GenomeTools* v1.5.9 *LTR_harverst*[77], and *LTR_retriver*[78] v2.9.0 were used to identify long terminal repeat (LTR) retrotransposons. We performed de novo searching to identify repeats by *RepeatModeler* v2.0.1 (http://www.repeatmasker.org/RepeatModeler/) and we classified the repeats into TEs superfamilies through *TEclass*[79] v2.1.3c. Finally, all TEs were merged into a repeat library that was used to annotate and mask sequences in the genome by *RepeatMasker* v4.1.1 (http://repeatmasker.org/).

## Gene annotation

Three approaches including ab initio prediction, homology-based prediction, and transcriptome-based assembly were used to predict gene structure in 100 genomes, including 14 wild, 41 local, 15 improved, and 30 genetic stocks. For homology-based prediction, *GeMoMa*[80] v1.6.10 was used to align the protein sequences of the reference silkworm, *Drosophila melanogaster*, *Apis mellifera*, and *Danaus plexippus* to the newly assembled silkworm genomes. For the transcriptome-based assembly, *STAR*[81] v2.7.3a was used to map the mRNA sequencing reads from 56 RNA samples, 15 strains, and 7 tissues (CNGB BioProject ID: CNP0001815, NCBI project ID: PRJNA262539, PRJNA264587, PRJNA407019) to the newly assembled genomes, *stringtie*[82] v2.1.4 was used to perform RNA assembly, and *PASA* v2.3.3[83] was used to predict open reading frames (ORFs). *augustus*[84] v3.4.0 with default parameters was used to perform ab initio gene prediction based on the training set obtained by *PASA*. Finally, *EVidenceModeler*[85] v1.1.1 (EVM) was used to produce an integrated gene set.

For each genome, information about gene function, motifs, and domains was obtained through homology searches against public databases including Swiss-Prot, NCBI non-redundant (NR) proteins, Kyoto Encyclopedia of Genes and Genomes (KEGG), Eukaryotic

Orthologous Groups (KOG), and Gene Ontology (GO). The GO terms of genes were identified using the *InterProScan*[86] v5.41-82.0 program with default parameters. For the other four annotations, the proteins of each genome were used as queries in *BLAST+*[87] v2.9.0 *BLASTP* (*e* value <1e-5) searches against Swiss-Prot, NR, KEGG, and KOG databases.

### ncRNA annotation

For the annotation of non-coding RNA (ncRNA), *tRNAscan-SE*[88] v2.0 was used to identify transfer RNAs (tRNAs), and *Infernal*[89] v1.1.3 *cmscan* was used to identify microRNAs and small nuclear RNAs (snRNAs) by searching against the *Rfam* database (http://rfam.xfam.org/). Then, *RNAmmer*[90] v1.2 was used to identify rRNAs.

### Pan-gene analysis

*OrthoFinder*[91] v2.3.7 with default parameters was used to cluster all genes of the 100 assembled genomes into orthologous groups that were classified into core, softcore, dispensable, and private genes.

### Identification, verification, and annotation of SVs

For ONT reads, SV calling based on the assembly and read-mapping approaches showed an overlap of 90% on average in genomes of tomato[7]. In this study, a similar pipeline was employed to identify structural variations for long-read sequenced samples. First, the ONT reads of each sample were aligned to the reference genome using *NGMLR*[92] v0.2.7. Then, *Sniffles*[92] v1.0.11 was used to call SVs. For filtering potentially spurious SVs, we first identified regions of the reference genome prone to producing false SV calls. To identify these regions, *SURVIVOR*[93] v1.0.3 was used to simulate ONT reads (~100× the genome coverage) from the reference genome and *Sniffles*[92] was used to call SVs. A total of 10 simulations was performed and the 10 VCF files containing structural variations were merged by *SURVIVOR*[93] (minimum distance = 1 kb, same SV type, and minimum SV length = 50 bp). The SVs located in these error-prone regions and their 2.5 kb flanking regions were filtered. SVs larger than 100 kb or with a "0/0" genotype were also removed. All SVs of the 545 genomes were merged using *Jasmine* v1.0.1 (min_support = 1, max_dist = 500, k_jaccard = 9, min_seq_id = 0.25, spec_len = 30,–run_iris).

To confirm the quality of the identified SVs, we first checked the reported SVs in our sequenced strains and counted how many SVs could be accurately identified. Next, we assayed 50 PAVs in the strains with and without SV by PCR with primer pairs flanking each SV followed by agarose gel electrophoresis. Finally, we mapped the short reads of each long-read sequencing strain to the pan-genome and estimated the precision, recall, and F1 score.

To identify repeat elements of SV sequences, the repeat library constructed from our 100 de novo assembled genomes was used to annotate and mask sequences in SVs using *RepeatMasker* v4.1.1 (http://repeatmasker.org/) with *RMBlast* v2.9.0-p2 (http://repeatmasker.org/RMBlast.html). The positional relationship between SVs and genes was annotated using *Vcfanno*[94] v0.3.2.

### Estimation of pan-gene plateau

Curves describing pan-gene number were fitted with the *nls* function in *R* according to previous studies[95–97]. The pan-gene number was estimated using the model $y = A + Be^{Cx}$. To evaluate pan-gene number in groups with a different number of genomes, we first selected 10 representative genomes based on phylogenetic relationships (Supplementary Fig. 1a) to carry out pan-gene number analysis. Subsequently, 10 genomes were added step by step up to 100 genomes in the following regression analyses. In total, 10 regression analyses were carried out. From the regression of 10 samples to the regression of 100 samples, the increment of gene number decreases gradually and was finally close to zero. The pan-gene increment fitted curve of each additional sample was also drawn (Fig. 3g, Supplementary Fig. 1c).

### Graph-based pan-genome construction

The linear reference genome and all PAVs of the 545 long-read sequencing genomes were used to construct a graph-based genome via *Vg toolkit*[98] v1.30.0. Based on this graph-based genome, we then used *Vg toolkit* pipeline considering the novel variants (using the augmented graph and gam) to call SVs in all the 537 samples that were only sequenced through NGS technology.

### RNA-seq analysis and qRT-PCR assay

The RNA sequencing data of each sample were mapped to the reference genome using *bowtie2*[99] v2.4.2. Gene read counts were calculated using *RSEM*[100] v1.3.3 and gene expression was normalized using the number of Fragments Per Kilobase per Million bases (FPKM). We performed qPCR using a Hieff® qPCR SYBR Green Master Mix (Yeasen) reaction system on the qTOWER3G system (Analytik Jena). The primers used in the qRT-PCR are listed in Supplementary Data 9.

### The impact of SVs on gene expression

The impact of SVs on gene expression was investigated using the previous approach[8]. Briefly, each SV-gene pair was defined by the distance (<5 kb) between SV and its related gene. We filtered these SV-gene pairs to retain the pairs that had one SV present in at least three and absent in at least three of the 14 samples with RNA-seq data. For each SV-gene pair, the samples were classified according to the presence or absence of SV groups. The differential expression genes between the two groups were compared using a student's *t* test, and *p* values underwent FDR correction using the Benjamini-Hochberg procedure.

### Identification of the SVs associated with domestication and improvement traits

The frequency of each SV between groups was compared using Fisher's exact test. The corrected *p* values (FDR) were calculated using the Benjamini-Hochberg procedure. The SVs with significantly divergent frequencies (FDR < 0.0001, fold change >2) in wild-local or local-improved comparisons were defined as domestication- or improve-associated SVs.

### CRISPR-Cas9 gene editing, transgenic overexpression, and RNA interference

For gene knockout, guide RNAs were designed with *CRISPRdirect*[101] v140413 and synthesized in BGI (China). The Cas9 protein was purchased from Invitrogen. Then, the mixture of Cas9 protein (0.5–0.8 ng) and guide RNA (5–8 ng) was injected into newly laid eggs by microinjection.

The transgenic vector piggyBac [3×P3-EGFP, Fib-H-BmE2F1-SV40] was constructed to overexpress *BmE2F1* gene in silk gland. The vector was injected with the helper plasmid pHA3PIG containing the piggyBac transposase sequence and the *B. mori* actin 3 promoter[102] into newly laid eggs by microinjection.

For RNA interference, short interfering RNAs were designed and synthesized in BGI (China). The siRNAs (250 μM, 0.5–0.75 μl/individual) were injected into the left side of the intersegment between the 7th and 8th segments of 3rd instar larvae. Then, conductive gel was placed at the injection site (left, positive pole) and control site (right, negative pole) and 15 V electric voltage was applied[103]. The sequences of sgRNA and siRNA used in this study are listed in Supplementary Data 9.

### Fineness measurement

Cocoons were reeled into single silk fibers. The silk fiber length and weight were measured. We then calculated fineness (*F*, dtex) of each cocoon using the formula:

$$F(\text{dtex}) = 10,000 \times \frac{\text{silk weight (g)}}{\text{silk length (m)}}. \tag{1}$$

## Reporting summary

Further information on research design is available in the Nature Research Reporting Summary linked to this article.

## Data availability

Raw data of the long-read sequencing and short-read sequencing (including RNA-seq and whole-genome sequencing) used in this study have been deposited in the CNGB Nucleotide Sequence Archive (CNSA) of China National GeneBank DataBase (CNGBdb, https://db.cngb.org) with BioProject ID CNP0001815 and also in the Genome Sequence Archive (https://ngdc.cncb.ac.cn/gsa/) with accession number CRA007878. All 545 genome assemblies, 100 genome annotations (gff files), pan-genome, and VCF files (SNP, SV) have also been deposited in the CNGBdb with BioProject ID CNP0002456. This study also analyzed data for four released wild silkworm genomes that are available in the Sequence Read Archive (SRA) database according to accession numbers DRX054041, DRX054040, ERS402904, ERS402902. All phenotypic data have been listed in the eighth column of Supplementary Data 1. Source data are provided with this paper.

## Code availability

The bioinformatic tools used in our study are all published or publicly available and are described in Methods.

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

## Acknowledgements

We thank Chao Su at Northwest A&F University, Bing Li at Soochow University, Keping Chen at Jiangsu University, Yuyin Chen at Zhejiang University, Weizheng Cui at Shandong Agriculture University, Chaobin Luo at Sericulture Institute of Guizhou, Yongqiang Wang at Zhejiang Academy of Agriculture Science, Wenfu Xiao, Gang Liu, and Yian Chen at Sericulture Research Institute of Sichuan, Zhanpeng Dong at Sericulture and Apicultural Research institute of Yunnan, Anjie Wang at Shandong Institute of Sericulture, Tao Fan at Sericulture Institute of Anhui, Lihui Bi at Sericulture Institute of Guangxi Zhuang Autonomous Region (GZAR), Fan Wu at Economy Crops Institute of Hubei, Yuxia Wang at Shanxi Research Institute of Sericulture, and Junwen Ai at the Sericultural Research Institute of Hunan for helping in the collection of silkworm strains. We also thank Desheng Zhang from Southwest University for the figure "Phenotypic diversity in silkworms". This work was supported by the National Natural Science Foundation of China (no. 31830094) to F.D., (no. U20A2058) to X.T., (no. 32002228) to LingLi Z., and (no. 32002229) to X.D., China Agriculture Research System of MOF and MARA (no. CARS-18-ZJ0102) to F.D., Creative Research Group of the Natural Science Foundation of Chongqing to F.D., and High-level Talents Program of Southwest University (no. SWURC2021001) to F.D.

## Author contributions

F.D., W.W., and X.T. conceived the project. F.D., Z.X., W.W., Z.T., and C.L. designed and supervised this project. F.D., H.H., X.T., Shubo L., Y.Y., C.L., T.G., Yahui Z., Jiangwen L., L.Z., Jinghou L., W.Z., Jiangbo S., S.H., S.W., Yunlong Z., Lei Z., Linli Z., L.C., Y.T., G.C., L.Y., R.Y., H.Q., Yanqun L., Y.P., Y.X., T.L., and A.X. collected samples for RNA-seq and Genome sequencing and performed phenotypic analysis. M.J.H., K.L., S.T., Yucheng L., Shubo L., Jianghong S., A.L., C.Z., Yanhong L., Z.W., W.H., J.X., T.F., Ye.Y., and J.W. performed genome assembly, SNP calling and phylogenetic analysis. M.J.H., K.L., Yanhong L., Shubo L., Jianghong S., A.L., and C.Z. performed genome annotation and pan-genome analysis. F.D., W.W., X.T., M.J.H., K.L., S.T., Y.L., Shubo L., Jianghong S., A.L., and C.Z. conducted the analysis of genomic variations related to complex traits. X.D., S.L., Q.G., B.Z., D.T., Y.Y., N.G., L.L., and Yaru L. conducted experiments. X.T., M.J.H., K.L., Shubo L., Jianghong S., A.L., C.Z., and Yucheng L. interpreted data and wrote the manuscript. F.D., W.W., Z.T., E.W., and A.M. revised the manuscript.

## Competing interests

The authors declare no competing interests.

## Additional information

Xiaoling Tong [1,2,18] ✉, Min-Jin Han[1,2,18], Kunpeng Lu [1,18], Shuaishuai Tai [3,18], Shubo Liang [1,18], Yucheng Liu[4,18], Hai Hu[1], Jianghong Shen[1], Anxing Long[2], Chengyu Zhan[1], Xin Ding[1], Shuo Liu[1], Qiang Gao[1], Bili Zhang[1], Linli Zhou[1,2], Duan Tan[1,2], Yajie Yuan[1], Nangkuo Guo[2], Yan-Hong Li[3], Zhangyan Wu[3], Lulu Liu[1], Chunlin Li[1], Yaru Lu[1], Tingting Gai[1], Yahui Zhang[1], Renkui Yang[5], Heying Qian[6], Yanqun Liu[7], Jiangwen Luo[1], Lu Zheng[1], Jinghou Lou[1], Yunwu Peng[8], Weidong Zuo[1,2], Jiangbo Song[1], Songzhen He[1], Songyuan Wu[1,2], Yunlong Zou[1,2], Lei Zhou[1], Lan Cheng[1,2], Yuxia Tang[1], Guotao Cheng[1,2], Lianwei Yuan[1], Weiming He [3], Jiabao Xu[3], Tao Fu[3], Yang Xiao[9], Ting Lei[5], Anying Xu [6], Ye Yin [3], Jian Wang[10,11], Antónia Monteiro [12,13], Eric Westhof[1,14], Cheng Lu [1] ✉, Zhixi Tian [4,15] ✉, Wen Wang [16,17] ✉, Zhonghuai Xiang [1] ✉ & Fangyin Dai [1,2] ✉

[1]State Key Laboratory of Silkworm Genome Biology, Institute of Sericulture and Systems Biology, Southwest University, Chongqing 400715, China. [2]Key Laboratory of Sericultural Biology and Genetic Breeding, Ministry of Agriculture and Rural Affairs, College of Sericulture, Textile and Biomass Sciences, Southwest University, Chongqing 400715, China. [3]BGI Genomics, BGI-Shenzhen, Shenzhen 518083, China. [4]State Key Laboratory of Plant Cell and Chromosome Engineering, Institute of Genetics and Developmental Biology, Innovation Academy for Seed Design, Chinese Academy of Sciences, Beijing 100101, China. [5]Chongqing Sericulture Science and Technology Research Institute, Chongqing 400715, China. [6]Jiangsu Key Laboratory of Sericulture Biology and Biotechnology, School of Biotechnology, Jiangsu University of Science and Technology, Zhenjiang, Jiangsu 212018, China. [7]College of Bioscience and Biotechnology,  Shenyang Agricultural University, Shenyang, Liaoning 111000, China. [8]Shaanxi Key Laboratory of Sericulture, Ankang University, Ankang, Shaanxi 710072, China. [9]Institute of Sericulture and Agricultural Products Processing, Guangdong Academy of Agricultural Sciences, Guangzhou, Guangdong 510000, China. [10]BGI-Shenzhen, Shenzhen 518083, China. [11]James D. Watson Institute of Genome Sciences, Hangzhou 310058, China. [12]Biological Sciences, National University of Singapore, 14 Science Drive 4, Singapore 117543, Singapore. [13]Science Division, Yale-NUS College, Singapore 138614, Singapore. [14]Architecture et Réactivité de l'ARN, Institut de Biologie Moléculaire et Cellulaire, UPR9002 CNRS, Université de Strasbourg, Strasbourg 67084, France. [15]University of Chinese Academy of Sciences, Beijing 100049, China. [16]School of Ecology and Environment, Northwestern Polytechnical University, Xi'an, Shaanxi 710072, China. [17]Kunming Institute of Zoology, Chinese Academy of Sciences, Kunming, Yunnan 650204, China. [18]These authors contributed equally: Xiaoling Tong, Min-Jin Han, Kunpeng Lu, Shuaishuai Tai, Shubo Liang, Yucheng Liu. ✉e-mail: xltong@swu.edu.cn; lucheng@swu.edu.cn; zxtian@genetics.ac.cn; wwang@mail.kiz.ac.cn; xbxzh@swu.edu.cn; fydai@swu.edu.cn

