## [Peer Review File · Nature Communications]

High-resolution silkworm pan-genome provides genetic insights into artificial selection and ecological adaptationReviewers' Comments:

Reviewer #1:

Remarks to the Author:

Review of "Sequencing of 1078 silkworm genomes provides genetic insights into commercially and ecologically important loci"

In this impressive study, Tong et al. investigated using both short and long reads to study the genetic variation of silkworm (*Bombyx mori*). The sampling is comprehensive and includes 1078 new genomes for both wild and domesticated populations. They ascertained many interesting properties of the pan-genome, with a particular focus on structural variations (SVs). The association between SV and silkworm domestication/breeding was investigated bioinformatically, and the authors provide four specific examples where using this new dataset enabled further genetic characterization of functionally important traits.

Major comments:

The strength of this manuscript is its comprehensive sampling, well-described pan-genome properties, and the functional investigation of genetic systems important for silkworm biology. The effort in generating such a broad dataset and incorporating various experiments to investigate silkworm domestication is well-worth the credit.

The paper is also clearly written and readily understandable.

In the past, many studies had the caveat of focusing mostly on SNP data while ignoring other types of genetic variation. This was mainly due to limitations of sequencing technologies (NGS, short read sequencing) that made the discovery of structural variants (particularly large SVs) difficult. By producing long read sequencing data and building a silkworm pangenome based on 545 individuals, this study was able to extensively characterize structural variation in this species.

However, my main concern is the way structural variations are interpreted in this manuscript: Some results on the contributions of SV and gene families to domestication and breeding seem to be overstretched, and likely require reinterpretation in a more conservative manner. Another related weakness is that the functional investigation of economic traits (silk quality) misleads readers to think that structural variation plays a role in altering the expression of these genes, while in fact experimental perturbations by CRISPR/RNAi only targeted coding sequences, so we do not know whether structural variations are relevant or not, hence whether the pangenome by itself adds much novel insight to the study of these traits. (The exception would be when studying diapause where the list of candidate loci was further narrowed down based on structural variations in exonic regions — in the case an exon deletion).

A separate issue is that there are many annoying abbreviations of specific terms, most of them apparently completely new for this paper. For very frequent terms (e.g., structural variation) it is understandable to use abbreviations, I think it would be unnecessary to shorten words such as "genetic linkage map", "domestication-associated genes", etc. Line 257-Line 259 is a particularly extreme example where these abbreviations considerably decrease the reader's comprehension:

"In the genomic and flanking (± 5 Kb) regions of IAGs, we identified 312 iSVs (improvement-associated SVs, showing significant divergent frequencies between improved and local groups) and most (99.7%) of these iSVs are within the PERRs of the IAGs, suggesting that the iSVs, like the dSVs, mainly alter PERR regions to modulate improvement-associated gene expression."

But there are many other examples. Many of these abbreviations are hardly used except in the text near by where they are introduced, and it would be much better simply to use the full wording; for

example, GLM is "genetic linkage map." It is used again, but it could easily be replaced simply by the term "map" which would be completely understandable in context...

The following comments are related to what I regard as somewhat overblown interpretations of structural variations. Because SVs are historically overlooked and the new dataset greatly improves our access to silkworm SVs, I recommend authors state these results in a more neutral tone, without stressing too much their importance to silkworm domestication or breeding, for reasons stated below:

L169-173: The change in frequency and types of gene families could be neutral, and the large frequency shift could be attributed to a large number of domesticated strains with relaxed selection, so that different strains fix different gene families (some of which could even be pseudogenes). There is no strong evidence from the data presented here that the newly identified genes have crucial roles in silkworm domestication and breeding – but one may argue in some way that domestication has influenced the fate of gene family evolution. For instance, in Fig 2j, the U-shaped distribution of variants frequency is intriguing. This could potentially be explained by domestication processes and the population structure (many small populations with restricted gene flow).

L221-223. Is this more/less than expected by chance, i.e. if these SVs were to be randomly placed along the genome? If more than expected by chance, could this suggest a greater accumulation of slightly deleterious mutations in domesticated silkworms?

L231: It is difficult to rule out whether changes in gene expression are caused by the appearance of a SV or by the presence of other regulatory alleles in those particular strains. Thus, it is not clear if SVs have a large impact on gene expression.

L244-245: Similarly, it is hard to rule out which SVs are associated with domestication and breeding purely by chance, so it's difficult to ascertain how many SVs have real impact on domestication. The fact that coding regions have much lower SV content can be understood by negative selection against deleterious mutations.

L259: The same issue as above.

L267-281: The presence of four SVs in the noncoding region does not preclude the possibility that BmE2F1's selection signal is attributed to changes in the coding sequence (no information on that in the main text). Extended Fig. 6a: Tajima's D does not look particularly different from the background in the gene BmE2F1, and is not very negative either, although Fst in Fig. 5a does show some elevated divergence. Is there any explanation for the behavior of Tajima's D?

L292-296: Again, the presence of two insertions in the fine-quality strains do not imply they are involved in the expression differences of this gene because CRSPR knockouts only affect coding sequences. An alternative hypothesis could be that an upstream regulatory gene shows allelic differences but no expression difference, and these different alleles alter the expression level of BmChit β -GlcNAcase. It is strange that knocking out BmChit β -GlcNAcase increases fineness, because in the previous sentence it suggests higher expression of this gene leads to finer silk. Is there any explanation for this?

L299-325: This particular study of diapause might be exempted from the caveats of interpretation, because a previous QTL scan has determined that diapause differences can be attributed to genetic variation surrounding chromosome 11 (11-55.89cM). Thus, if the 747 bp deletion is the only variation surrounding gene BmTret1-like, between the two phenotypes, the conclusion will be quite solid.

L326-358: I have no strong opinion on this study, as there are prior results indicating the function of L and L^c locus, and the role of SVs is not overstated here.

Minor Comments/Corrections:

L57. "We find that silkworm population harbors extremely variable genomes [...]" . In what sense? Size variation, structural variants, SNPs? Quantify it!

I would be intrigued to know HOW much larger the pangenome is than a typical sample genome. It seems it might be quite high, with ~500 insertions, and an average of length of 10 kb, giving an average of an extra 5 Mb. But I had the impression that my rough calculation could be an enormous underestimate.

L87. "We identified the SVs and their influencing genes underlying domestication and breeding processes in silkworm". Consider rephrasing.

L97. It would be good at some point in the manuscript to have a brief description of what of each of these categories (local strains, improved varieties, genetic stocks and wild silkworms) are and how they are related to each other.

L106-107. "[...] key determinants of the population structure of silkworms are artificial selection and [...]". The distinction on PC1 between wild and domesticated populations is not necessarily explained by artificial selection alone but by the domestication process (which includes drift and relaxed purifying selection in neutral regions of the genome, not related to domestication traits).

L107. "The resources are divided [...]". What does resources here mean? Probably there's a better word.

L105-L110. These lines could be probably simplified to pass the main message which is that PC1 splits individuals into wild and domestic groups, while PC2 further divides individuals into groups based on their geographic origin.

L110-112. I couldn't understand what the authors meant in the second part of this sentence. Please clarify.

L113. "The results generated four major subgroups". Perhaps it would be better to rephrase to something as "The results show the existence of four major subgroups", since the groups already exist, and the analyses only help describe them.

L130. "To reach an overview of genomic content in silkworms". Perhaps replace "reach" by "give" or "present".

L135-138. It could be perhaps interesting to give a figure of how many of these assemblies are chromosome level assemblies or have at least some entire chromosomes in a single scaffold.

L138-139 – Here do the authors mean the percentage of "complete single copy BUSCOs" or does it include also duplicated and fragmented BUSCOs? Also, is the mapping ratio the percentage of all mapped reads or uniquely mapped reads?

L159: "wildest" should be "widest"?

L172. Change from "Among which" to "Among these".

L180. Should also mention figure 3b.

L235-236. Perhaps it would be worth mentioning on what basis these candidates were identified (i.e. the intersection of FST, Tajima's D and XP-CLR).

Fig 1: CHN-I and JPN-I cannot be found on the spatial map. Does that mean they are only present in laboratories?

Also, I felt there should be a scale on the y axis of Fig. 1b -- it would be interesting to get some idea how divergent the wild populations are from the domesticated populations. You could also use "Neighbor-joining tree" instead of phylogenetic tree, since this is not really a phylogeny, but a distance-based representation of strains within a species.

Fig 3d: Many rings lack a y-axis (or description of the range of y-axis)

Fig 3a & 3c: Both figures have x-axis labeled as "Number of samples", but if I understand correctly, 3a means "each individual sample", and 3c means "the number of included samples".

Reviewer #2:

Remarks to the Author:

GENERAL COMMENTS

Key Results

This manuscript reports construction of a pan-genome for the domesticated silkworm, *Bombyx mori*, and its nearest wild ancestor, *B. mandarina*, starting with de novo next-generation sequences (NGS) of more than 1000 silkworms from well-maintained, diverse collections of stocks used for practical breeding (so-called "local" or genetically "improved" strains), and maintenance and analysis of genetic mutants (called "genetic stocks" carrying documented morphological and biochemical mutations), and from wild populations of *B. mandarina* collected in China. The authors conducted principle component analysis on data for single nucleotide polymorphism (SNPs) from this large-scale NGS project to investigate determinants of population structure which have a bearing on the history of silkworm domestication and geographic origins of modern strains. From the phylogenetic tree for these data the authors then chose 545 representative strains to construct a high fidelity deep coverage pan-genome using nanopore long-read sequencing. The report includes standard parameters to assess completeness of a genome assembly (e.g., average read depth, length and total genome size, fraction of repeated sequences, percent of well-conserved single copy orthologs or BUSCO, etc.), and to assess coverage and completeness of a pan-genome, such as minimum number of long-read genomes needed to fully represent the source populations (determined to be 100 in this case), its basic gene information content (such as number of genes and gene families, assignment of sequences to "core" (shared by all samples), softcore (shared by >90% but <100% samples), dispensable (shared by >1 but ≤ 90% samples) and private (1% samples) categories, the properties and numbers of structural variants (SVs) and transposable elements (TEs) relative to the published reference genome for silkworm and to pan-genomes for *Drosophila melanogaster* and human, and so on. In their analysis and interpretation of these and other data the authors consistently made the distinction between domesticated, genetically improved, and wild silkworms, emphasizing the relevant impact of domestication and subsequent stock development and maintenance, important subjects for silkworm scientists in the data.

Of broad interest is the authors' treatment of the "Impact of SVs on genes," in which they report the proximity of indels, inserts, and so on in genes' flanking and coding regions and introns (defined as "potential expression regulatory regions" or PERRS), assign GO terms, and use RNA-seq to measure expression in relatively small but apparently effective number (14) of silkworm strains with or without a target SV. Interestingly, roughly 9% showed differential expression in at least one of 6 tissues used for this study. Given how generalized these data sets are, it is encouraging that the authors were able to find published evidence for predicted SV involvement in 9 experimentally characterized silkworm

mutants included in the pan-genome itself (Suppl. Table 6a). My guess is that a more extensive review of the literature will find a similar association of SVs in most (all?) published silkworm mutations analyzed by laborious traditional positional cloning methods. Being able to initiate a search for more subtle and likely polygenic mutations using the fruits of the kind of analysis reported here should enable finding many more SVs affecting expression of important genes in a kind of reverse functional analysis. The authors demonstrated the potential strength of this approach by defining more than 400 "domestication-associated genes" in a comparison of SV-associations between wild and domesticated pan-genomes, more than half of which are newly described compared to previous studies. The authors took a similar approach to identify potential "improvement-associated SVs" by comparing pan-genomes of improved Chinese (CHN-I) and Japanese (JPN-I) to the corresponding local strains, again identifying many potential new targets for further analysis. Readers will be able to peruse the extended tables associated with these studies to see whether their favorite genes meet criteria as candidates for their own research.

The authors provide 6 concrete examples illustrating the success of this pan-genome approach for functional genomics by newly identifying 2 genes affecting key sericultural traits (silk yield and fineness), and 1 each affecting embryonic diapause (using a streamlined positional cloning approach on a well-known homozygous diapause mutant), and a larval body marking. In all cases they confirmed the identifications with targeted gene knockouts and in one case ectopic expression. Although the report of these findings could probably stand on their own in a separate publication, presenting it here stands as a strong, concrete illustration of the power of the new pan-genome for important applications in silkworm biology, genetics, and functional genomics.

Validity and robustness of data and analytic approach

The report presents extensive data to support and validate the construction and composition of a silkworm pan-genome, its properties, and potential applications. I do not have enough technical expertise in bioinformatics or statistics to evaluate these aspects of it effectively so I will not comment further on them except in general terms in other areas of this report.

Significance

This project addresses many major issues which are relevant and important for silkworm biology. As a central theme the focus on the molecular genetic (genomic) basis of domestication, in which traits derived from *B. mandarina* were ultimately fixed by selection, is estimated to have begun more than 5000 years ago. The ongoing selection for traits favorable to sericulture and capture of spontaneous and induced mutations which arose over the years have resulted in a highly diverse collection of genetic material which has yet to be effectively mined for a heritage which is unique among insects. As noted in the report, some of these characters were studied previously but with relatively little enlightenment regarding their molecular basis and thus remain unexplained. Here the authors provide substantial evidence for the idea that a pan-genomic approach can not only answer longstanding questions about the history of sericulture but, equally important, uncover many more subtle and genetically complex traits which promise to open up new areas for future study in silkworms and other insects.

Clarity, Context and Suggested improvements

For the most part I found the report to be relatively complete, clear, and well-written, with a minimum of common misuses of fine points of English grammar (authors please see some detailed comments below and written directly on the manuscript). A relatively minor exception is that, whereas the overall manuscript's organization of Introduction, Results, Methods, and Discussion is as expected, with mostly necessary and appropriate information contained in those sections, I found some crossing of those boundaries with information I think can be presented more effectively in a different section. This occurred mostly in Methods where, after describing a procedure or protocol, the authors summarized

(or repeated) the findings and conclusions from that part of the project. Strictly speaking, I believe most of that kind of information belongs in Results (not Methods) where it is essential for readers to know and understand the nature and quality of the authors' findings. Further, not only may finding it reported under Methods confound readers' expectations, it may also be missed by readers who (for various reasons) skim or skip reading the Methods section.

Additional General and Specific Comments about content are listed separately below under each section of the manuscript.

General Comments About Writing And Expression

Although the manuscript as a whole reads well (as noted above), I urge the authors to be consistent in their use of verb tense. In my understanding the general policy for scientific journals is to report new results in a manuscript in the past tense (e.g., the indicating authors "did" something and "found X, Y, Z"). In contrast, published results are reported in the present tense as "true" or existing "facts." And hypotheses and conclusions are (perhaps confusingly?) written in the present tense as being newly reported. Here the authors usually report their findings in the present tense, which is not consistent with these "rules" but okay with me if also okay with the journal. However, in a few places the authors use past tense to report their new results or observations. I have marked these cases in the manuscript when I noticed the difference but did not try to do this consistently and leave changes for overall consistency to the authors (or copy editors?).

In several places I recommend deleting "respectively" as unnecessary where the word order of items in successive lists within a single sentence is obvious. Or (at least in one case) there are no successive lists. Again, specific examples of this are marked on the manuscript and/or noted below.

Suggestions for Content and Specific Line-By-Line Comments by Manuscript Section

Introduction

No specific (line) comments here. I found this section to be relevant and concise. Although the authors might consider adding a somewhat more general article (a summary or review) on the current status of pan-genomics in addition to those cited, such as Golicz et al. (2020) Trends in Genetics <https://doi.org/10.1016/j.tig.2019.11.006>. And possibly consider citing a recent bioRxiv prepublication reporting a 3-species pan-genome for members of the Heliconius butterfly clade designed to examine the evolution of chromatin accessibility (Ruggieri et al., <https://www.biorxiv.org/content/10.1101/2022.04.14.488334v1>). The appearance of this and the present publication suggests a pan-genome approach will soon be used more often for moths and butterflies, especially given a rapid increase in the number of reference genomes for these clades (e.g., see Ellis et al., <https://doi.org/10.1093/gigascience/giab041> and Darwin Tree of Life Project, among other possible sources).

RESULTS

General comments

I found most of the content of Results to be of interest and reasonable, with one exception, notably, the authors' treatment of multigene families. At first I found somewhat startling their assertions of having found so many multigene families – indeed, a great many more than in the last published reference genome for silkworm, which is reasonably complete. The authors identify sequences defined here as members of multigene families by standard bioinformatic criteria and organize and classify them in what I assume are standard ways used in genomics and pan-genomics such as annotation using various resources (NR, GO, KEGG, and KOG terms) with resulting amino acid sequences, evidence for expression via RNA-seq (carried out by the authors) reported as FPKM values, and

assignment to core, softcore, and dispensable categories based on the degree of sequence conservation among the strains comprising the pan-genome. Inspection of supplementary Table 4a which contains these data revealed a wide assortment of types of sequences, including transposable elements, viruses, reverse transcriptases, untranslated RNAs, etc. etc., together with genes coding for well-studied kinds and classes of structural proteins. Although this somewhat simplified approach to characterizing and classifying multigene families may be reasonable in terms of "pure" bioinformatics where annotations rely on somewhat arbitrary criteria for cataloguing and grouping sequences, it leaves out an important consideration of gene function, which results in vastly different structural constraints and consequently evolutionary properties, depending on whether a sequence encodes a viral remnant, a transposable element, a reverse transcriptase, a noncoding RNA, or a structural protein, to name just a few examples. The resulting number of family members, degree of sequence identity, and inferred evolution of these different types of "genes" will be vastly different, and, perhaps, should not be considered and reported using the same general criteria.

A specific example of my problem with this pooling of data for all types of "multigene families" is the result of a search for "chorion," a multigene family for a class of structural proteins with which I am familiar, where I found 34 sequences under the column of "NR" (NCBI annotation). This corresponds well to the reported number of chorion genes (and proteins). However, here 27 of them are marked "y" for "newly identified," and only 7 are listed as "n" or not newly identified. Of the so-called "newly identified" chorion proteins, 4 are listed as "softcore" or moderately conserved, and the remainder as "dispensable" (which should be spelled "dispensable"), meaning not very well conserved; whereas roughly half (3) of the so-called previously identified chorion proteins are listed as "softcore" and half (4) as "dispensable." The assignments to these categories and their inferred evolutionary histories simply don't jibe with the well-established number, family member distribution and conservation of chorion proteins which are well-characterized at protein and DNA levels and have been examined in depth in two evolutionarily diverged silkworm strains, the genome reference strain Daizo and the European strain 703. This misdirection could simply be a result of the over-general way the gene-finding algorithms assign identity to families with structural members which are diverse at a micro-level but, nevertheless, maintain overlapping or even identical functions, which indicates they are basically well-conserved. In other words, it is unlikely that the 27 labeled here as "newly identified" are actually "new;" it's just that their sequences have diverged enough among each other (and from the original reference strain) to be defined as "new" by the limited bioinformatic criteria used here. Although I did not investigate this issue further, I am certain readers will have similar reservations about the treatment of other well-characterized multigene families which encode structural proteins that have been important research subjects in silkworm and other insects, such as cuticle proteins, detoxification enzymes, neural peptides, and so on. These observations suggest to me the authors should consider adding to the Results (perhaps in supplementary Table 4 or in another supplementary table) some sub-grouping of gene family members into at least rough functional categories. And add to the Discussion some information (and perhaps caveats) about the nature and identities of sequences listed as "multigene families" in this report.

Specific Comments

Lines 248-250: misuse of "respectively" (please see comments above; "respectively" is not used when the word order is clear)

248 We identified 126 (CHN-I)

and 116 (JPN-I) improvement-associated regions (IARs) containing 106 and 92

improvement-associated genes (IAGs), respectively (Fig. 4c, Supplementary Table 8).

Lines 228-231 leave me with some confusion. $1560/2396 = 65\%$ - so I do not understand where the value of 9.2% comes from

228 Finally, 1,560 genes

in 2,396 SV-gene pairs (9.2%) were found to be differentially expressed (FDR < 230 0.001) in at least one tissue between strains with and without corresponding SV

(Extended Data Fig. 4h), indicating a large impact of SVs on gene expression.

Lines 235-237: Authors, please indicate here how you define or describe the term “domestication-associated” genes (DAGs). I believe you did this later in the manuscript but it would be better to do it the first time the term appears.

desirable traits in silkworm. We identified 468 (2.8% of the whole-genome genes)

domestication-associated genes (DAGs) (Fig. 4a, Supplementary Table 8), containing

264 newly identified DAGs compared with previous studies^{19, 36, 37}.

Lines 284-286: Please define or explain SX, CF, XF, and QB which appear here for the first time in the MS. It is possible the authors mean to indicate SX and CF are associated with fine silk, and XF and QB are associated with coarse silk. However, the way parentheses () are used here confuses rather than clarifies the situation, at least for me.

284 Here we performed RNA-seq of the silk press in fine silk (SX

285 (BomP174), CF (BomP79)) and coarse silk (XF (BomP154), QB (BomP31)) strains

286 (Fig. 5e, Extended Data Fig. 7a)

Lines 356-358 I was confused at first by the authors’ use of “mapping cloning.” I believe a better term is “map-based” cloning.

356 These results reveal that large and complex SVs in L alleles, which cannot be obtained

357 by mapping cloning, affect the expression pattern of Wnt1 and result in twin-spot

markings.

DISCUSSION

General Comments

In addition to the suggestion above under Results and one below under Methods, Perhaps it would be helpful to readers to suggest other areas for future study which are associated with silkworm domestication and have been relatively refractory to a “classic” trait-based genetic mapping/sequencing approach. For example, differences in *B. mori* behavior relative to *B. mandarina*, such as larval ability to withstand crowding and handling, relatively docile feeding (lacking a strong drive for finding food), and loss of flight ability. Judicious choice of these could lead to new insights in other Lepidoptera or even in other kinds of insects.

Specific Comments

Lines 378-380: I question the breadth of this statement with reference to its use of “various species.” It just seems a little too vague and general. Specifically, I question the extent to which findings in silkworms will help in the understanding the domestication of any other ANIMAL species (plants less unlikely) except perhaps for insects, but probably not mammals or avians, which are the main species I believe we can say have been “domesticated”. And given that (in my understanding) no insect except silkworm is truly domesticated, i.e., having been so strongly selected for traits of value to humans that it is, effectively, significantly different from its nearest wild ancestor (and therefore merits being called a separate species) and fully dependent on us for survival, this statement would be more convincing if focussed (at least) on insects, or even, possibly, only on Lepidoptera and, perhaps, honeybees.

The analysis of the functions of these DAGs and IAGs will reveal the genetic basis of

artificial selection and provide improvement targets, as well as help the understanding

of the common genetic mechanisms underlying domestication of various species.

Lines 384-388: I find this statement to be clear enough but it is unnecessarily repetitious (stating silkworm economic traits twice in two short sentences). Further, it could be expanded to suggest broader applications. I suggest changing it to something like the following (perhaps leaving out the

underlined phrases as overstated and unnecessary since they refer to a continuing theme of the report):

“Furthermore, our use of a large-scale pan-genome to decipher two genes that control important economic traits in silkworms may also be used to reveal genetic mechanisms and traits associated with the survival of wild populations and evolution of new species under strong natural selection by human and non-human factors.”

Furthermore, we

deciphered two genes (BmE2F1 and BmChit β -GlcNAcase) that control important
economic traits in silkworms relevant to silk yield and fineness using large-scale pan-
genome. These findings have significance for improving economic traits of silkworm
varieties.

Methods

Lines 429-432: Although this statement contains basic information about the experimental design in terms of numbers of samples of various categories the authors used in this research, it also contains a specific comparison with previous studies which I suggest is more appropriately reported in a Discussion section than in Methods.

of their geographic distributions in China. We have a larger sample size and a wider
geographic distribution of sample set compared with previous publications that
contained 40 (11 wild silkworms and 29 domestic silkworms) and 144 (seven wild
silkworms and 137 domestic silkworms) strains in 200936 and 201819.

Line 469: Authors: Please clarify or explain what you mean by the term “regular.” Perhaps replace with a more technical descriptive term.

469 kb de novo regular library of each sample was used to sequence on PromethION

Lines 531-536: These lines describe new data from the study. For reasons noted above I believe this information belongs in the Results section and should be removed from Methods. I have a similar concern about information reported in lines 540-543, 576-581, 609-612, 616-625, and 692-695 which are underlined in the text but not extracted here.

For domestication, 468 genes were identified as potential
domestication-associated genes (DAGs). Comparing to previous studies^{19, 36, 37}, we
newly identified 264 DAGs in our extended panels of wild and domestic silkworms.
For improvement, we identified 189 improvement-associated genes (IAGs) containing
nine genes shared by CHN-I and JPN-I. 185 of those genes are newly identified
compared with IAGs in the study of Xiang et al¹⁹.

Line 703: Shouldn't the authors list here the helper plasmid used in the CRISPR-cas9 procedure?

701 The transgenic vector piggyBac [3×P3-EGFP, Fib-H-BmE2F1-SV40] was
702 constructed to over express BmE2F1 gene in silk gland. The vector was injected with
703 the helper plasmid into newly laid eggs by microinjection.

(signed) Marian R. Goldsmith

**Response to Reviewer #1:**

Review of “Sequencing of 1078 silkworm genomes provides genetic insights into commercially and
ecologically important loci”

In this impressive study, Tong et al. investigated using both short and long reads to study the genetic
variation of silkworm (*Bombyx mori*). The sampling is comprehensive and includes 1,078 new genomes
for both wild and domesticated populations. They ascertained many interesting properties of the pan-
genome, with a particular focus on structural variations (SVs). The association between SV and silkworm
domestication/breeding was investigated bioinformatically, and the authors provide four specific examples
where using this new dataset enabled further genetic characterization of functionally important traits.

**Response:** Thank you very much for reviewing our manuscript and providing valuable comments. We
have done a point-by-point reply for your comments below.

Major comments:

The strength of this manuscript is its comprehensive sampling, well-described pan-genome properties,
and the functional investigation of genetic systems important for silkworm biology. The effort in generating
such a broad dataset and incorporating various experiments to investigate silkworm domestication is well-
worth the credit.

The paper is also clearly written and readily understandable.

In the past, many studies had the caveat of focusing mostly on SNP data while ignoring other types of
genetic variation. This was mainly due to limitations of sequencing technologies (NGS, short read
sequencing) that made the discovery of structural variants (particularly large SVs) difficult. By producing
long read sequencing data and building a silkworm pangenome based on 545 individuals, this study was
able to extensively characterize structural variation in this species.

**Response:** Thank you very much for your positive comments.

However, my main concern is the way structural variations are interpreted in this manuscript: Some results
on the contributions of SV and gene families to domestication and breeding seem to be overstretched,
and likely require reinterpretation in a more conservative manner. Another related weakness is that the
functional investigation of economic traits (silk quality) misleads readers to think that structural variation
plays a role in altering the expression of these genes, while in fact experimental perturbations by
CRISPR/RNAi only targeted coding sequences, so we do not know whether structural variations are
relevant or not, hence whether the pangenome by itself adds much novel insight to the study of these
traits. (The exception would be when studying diapause where the list of candidate loci was further
narrowed down based on structural variations in exonic regions — in the case an exon deletion).

**Response:** Thank you for raising these points. We absolutely agree with you. We modified the
interpretation of the contributions of SV and genes to domestication and breeding in more conservative
manner. In the revised manuscript, please see Lines 87-88, we modified the sentence as “ We identified
hundreds of SVs and genes potentially underlying domestication and breeding of silkworm. ”. Lines 178-
179, the sentence was changed into “Among these genes, 72% and 82% were newly identified, indicating

that the newly identified genes have potential roles in silkworm domestication and breeding.”. We deleted
“indicating a large impact of SVs on gene expression” , “These results suggest that SVs impact
domestication-associated genes in a transcriptional regulatory way.” and “suggesting that the iSVs, like
the dSVs, mainly alter PERR regions to modulate improvement-associated gene expression” in the
revised manuscript.

Indeed, if these structural variations could be precisely edited or corrected, the association between the
SVs and phenotype should be more convictive. We performed CRISPR-Cas9/RNAi mediated
knockout/knock down for four genes, including *BmE2F1*, *BmChit β-GlcNAcase*, *BmTret1-like*, and *Wnt1*.
Multiple SVs or super-size (6.2 kb-271 kb) SVs were detected to be related to these genes in their cis-
regulatory regions or introns. It is not easy to precisely edit those SVs synchronously or edit such large
SVs right now. So, we assessed the gene expression level and edited the coding genes to test the role of
these genes in phenotype determination to provide the first piece of functional evidence. Now, we didn't
know whether the differential expression of *BmChit β-GlcNAcase* between fine silk strains and coarse silk
strains was caused by these SVs. However, these SVs are unique to the fine silk strains relative to the
coarse silk strains, we speculate that these SVs should be related to or tightly linked with the causal
variations of fineness diversity. More experiments are required to test the role of these SVs.

The pan-genome by itself indeed adds some novel insight to the study of these traits. Although dozens
of silkworm mutants have been identified through traditional positional cloning and NGS data, the mapping
cloning is time-consuming and inefficient to decode the traits caused by complex structure variation,
variation in intergenic regions, and the new genes absent in the reference genome. Here, the long-read
based pan-genome enable us to efficiently decipher the genetic variations related to traits. For instance,
these two large SVs (34 kb duplication and 109 kb insertion) in *L* strain were not detected in the previous
study by map-based cloning and short read sequence but were captured in our data.

A separate issue is that there are many annoying abbreviations of specific terms, most of them apparently
completely new for this paper. For very frequent terms (e.g., structural variation) it is understandable to
use abbreviations, I think it would be unnecessary to shorten words such as “genetic linkage map”,
“domestication-associated genes”, etc. Line 257-Line 259 is a particularly extreme example where these
abbreviations considerably decrease the readers comprehension:

"In the genomic and flanking (± 5 Kb) regions of IAGs, we identified 312 iSVs (improvement-associated
SVs, showing significant divergent frequencies between improved and local groups) and most (99.7%) of
these iSVs are within the PERRs of the IAGs, suggesting that the iSVs, like the dSVs, mainly alter PERR
regions to modulate improvement-associated gene expression."

But there are many other examples. Many of these abbreviations are hardly used except in the text near
by where they are introduced, and it would be much better simply to use the full wording; for example,
GLM is "genetic linkage map." It is used again, but it could easily be replaced simply by the term "map"
which would be completely understandable in context...

**Response:** Thank you for raising these points. We are sorry to confuse you with too many
abbreviations. According to your suggestion, we checked the whole manuscript and changed these
abbreviations including GLM, dAGs, dSVs, IAGs, iSVs, IARs, PERRs to “genetic linkage map”,
“domestication-associated genes”, “domestication-associated SVs”, “improvement-associated genes”,
“improvement-associated SVs”, “improvement-associated regions” , and “potential expression
regulatory regions”.

The following comments are related to what I regard as somewhat overblown interpretations of
structural variations. Because SVs are historically overlooked and the new dataset greatly improves our
access to silkworm SVs, I recommend authors state these results in a more neutral tone, without
stressing too much their importance to silkworm domestication or breeding, for reasons stated below:
L169-173: The change in frequency and types of gene families could be neutral, and the large frequency
shift could be attributed to a large number of domesticated strains with relaxed selection, so that
different strains fix different gene families (some of which could even be pseudogenes). There is no
strong evidence from the data presented here that the newly identified genes have crucial roles in
silkworm domestication and breeding – but one may argue in some way that domestication has
influenced the fate of gene family evolution. For instance, in Fig 2], the U-shaped distribution of variants
frequency is intriguing. This could potentially be explained by domestication processes and the
population structure (many small populations with restricted gene flow).

**Response:** Thank you for your comment and suggestion. We agree with you. Relaxed selection,
domestication/breeding process and population structure can cause frequency change of genes. Thus,
for more rigorous, we revised our conclusion to “Among these genes, 72% and 82% were newly identified
genes, indicating that the newly identified genes have potential roles in silkworm domestication and
breeding.” Please see line 178-179 in the revised manuscript.

L221-223. Is this more/less than expected by chance, i.e. if these SVs were to be randomly placed along
the genome? If more than expected by chance, could this suggest a greater accumulation of slightly
deleterious mutations in domesticated silkworms?

**Response:** We identified ~55.15% of total SVs (1,892,838 SVs) in the potential expression regulatory
and coding gene regions (257.11 Mb) that occupied ~55.85% of the reference genome size (460.33 Mb).
The proportion (55.15%) of these SVs in the total SVs is close to the proportion (55.85%) of these regions
(potential expression regulatory and coding gene regions) in the genome. Thus, the observed 55% SVs
in potential expression regulatory regions and coding sequence of reference genes is understandable and
consistent with the expected.

L231: It is difficult to rule out whether changes in gene expression are caused by the appearance of a SV
or by the presence of other regulatory alleles in those particular strains. Thus, it is not clear if SVs have a
large impact on gene expression.

**Response:** We agree with you. Based on our results, we are not sure whether changes in gene
expression are caused by SV or by the presence of other regulatory alleles, which are difficult to figure
out at present. We thus have deleted “indicating a large impact of SVs on gene expression” in the revised
manuscript.

L244-245: Similarly, it is hard to rule out which SVs are associated with domestication and breeding purely
by chance, so it's difficult to ascertain how many SVs have real impact on domestication. The fact that
coding regions have much lower SV content can be understood by negative selection against deleterious
mutations.

L259: The same issue as above.

**Response:** Thank you for your comment. In our study, we compared allele frequencies of SVs between
wild and local populations (or between local and improved populations). The SVs with $FDR < 0.001$ and
132 fold change > 2 are deemed as the SVs potentially associated with domestication or improvement process.

We agree with you that there still many works to be done to give direct evidences of relationships between
 SVs and gene expression, as well as their effects on domestication/improvement process. Thus, we
 deleted "These results suggest that SVs impact domestication-associated genes in a transcriptional
 regulatory way." and "suggesting that the iSVs, like the dSVs, mainly alter PERR regions to modulate
 improvement-associated gene expression" in the revised manuscript.

L267-281: The presence of four SVs in the noncoding region does not preclude the possibility that
 BmE2F1's selection signal is attributed to changes in the coding sequence (no information on that in the
 main text). Extended Fig. 6a: Tajima's D does not look particularly different from the background in the
 gene BmE2F1, and is not very negative either, although Fst in Fig. 5a does show some elevated
 divergence. Is there any explanation for the behavior of Tajima's D?

**Response:** Thank you for your comment. We detected four SVs that are differentially distributed between
 local and CHN-I silkworms in the flanking regions and intron of *BmE2F1*. According to your suggestion,
 we also analyzed SNPs and Indels in the coding regions of *BmE2F1* (as shown in Table 1 and Table 2
 below). We found that the allele frequencies of two missense SNPs in CDS and two SNPs in the 3' UTR
 of *BmE2F1* are significantly divergent between local and CHN-I silkworms (Table 1). The two missense
 SNPs caused the alteration of two amino acid located at 349 and 398 sites. More experiment need to be
 done to confirm the influence of these SVs and SNPs on the expression or function of *BmE2F1*.

The Tajima's D and F_{ST} are two statistic method of identifying selective sweep based on different signature
 of genetic variation. The F_{ST} is a measure of differences in allele frequencies between populations. The
 Tajima's D is a measure of reduction in genetic diversity. The relatively modest behavior of Tajima's D in
 the genomic region of *BmE2F1* reflects the reduction of genetic diversity that is not very dramatic. However,
 in a genomic region of the 5' of *BmE2F1*, the Tajima's D statistic is in the threshold of lowest 5% Tajima's
 $D_{improvement}$ (and Tajima's $D_{improvement} < D_{local}$), implying positive selection in this region.

Table 1. SNPs in the exons of *BmE2F1*

Chr	Position	Exon ID	Ref	Alt	Local group		CHN-I group		Chi-square	P value	Variant type
					Allele frequency	Sample number	Allele frequency	Sample number			
Chr15	13344318	2	C	T	0.02439	205	0	105	1.293	0.256	missense
Chr15	13353071	3	C	T	0.009756	205	0	105	1.661	0.197	synonymous
Chr15	13353137	3	G	C	0.256098	205	0.890476	105	109.62	<0.0001	synonymous
Chr15	13355780	5	A	C	0.002439	205	0	105	1.293	0.256	synonymous
Chr15	13356935	6	T	C	0.053659	205	0.0190476	105	1.298	0.254	synonymous
Chr15	13356950	6	A	T	0.478049	205	0.0333333	105	63.864	<0.0001	synonymous
Chr15	13356971	6	T	G	0.002439	205	0	105	1.293	0.256	synonymous
Chr15	13356980	6	A	G	0.004878	205	0	105	0.829	0.363	synonymous
Chr15	13358006	8	C	T	0.163415	205	0.861905	105	141.719	<0.0001	missense
Chr15	13358010	8	A	G	0.002439	205	0	105	1.293	0.256	synonymous
Chr15	13358152	8	G	A	0.636585	205	0.9	105	23.989	<0.0001	missense
Chr15	13358169	8	G	A	0.004878	205	0	105	0.829	0.363	synonymous
Chr15	13358894	9	C	T	0.041463	205	0.0190476	105	0.363	0.547	synonymous
Chr15	13358909	9	C	T	0.039024	205	0	105	2.797	0.094	synonymous
Chr15	13358957	9	C	G	0.056098	205	0.0190476	105	1.679	0.195	synonymous
Chr15	13358972	9	G	A	0.056098	205	0.0190476	105	1.679	0.195	synonymous

Chr15	13358978	9	G	C	0.056098	205	0.0190476	105	1.679	0.195	synonymous
Chr15	13367307	10	C	T	0.134146	205	0.0857143	105	1.431	0.232	3_prime_UTR
Chr15	13367311	10	G	C	0.131707	205	0.0857143	105	1.431	0.232	3_prime_UTR
Chr15	13367373	10	A	C	0.42439	205	0.947619	105	77.77	<0.0001	3_prime_UTR
Chr15	13367418	10	G	A	0.419512	205	0.947619	105	79.031	<0.0001	3_prime_UTR
Chr15	13367436	10	T	C	0.182927	205	0.104762	105	3.39	0.066	3_prime_UTR
Chr15	13367445	10	G	T	0.182927	205	0.104762	105	3.39	0.066	3_prime_UTR

Table 2. Indels in the exons of *BmE2F1*

Chr	Position	Exon ID	Ref	Alt	Local group		CHN-I group		Chi-square	P value	Variant type
					Allele frequency	Sample number	Allele frequency	Sample number			
Chr15	13358065	8	C	CTCG	0.0390244	205	0	105	2.797	0.094	Disruptive inframe insertion
Chr15	13367399	10	T	TTTC	0.00487805	205	0.00952381	105	0.221	0.638	3 prime UTR variant
Chr15	13367457	10	TG	T	0.00243902	205	0	105	1.293	0.256	Splice region variant

L292-296: Again, the presence of two insertions in the fine-quality strains do not imply they are involved in the expression differences of this gene because CRSPR knockouts only affect coding sequences. An alternative hypothesis could be that an upstream regulatory gene shows allelic differences but no expression difference, and these different alleles alter the expression level of *BmChit β-GlcNAcase*. It is strange that knocking out *BmChit β-GlcNAcase* increases fineness, because in the previous sentence it suggests higher expression of this gene leads to finer silk. Is there any explanation for this?

Response: Thank you for raising this point. To identify the gene responsible for silk fineness, we first identified the differentially expressed genes between fine silk and coarse silk trains. Among these differential expression genes, we found that *BmChit β-GlcNAcase* gene harbors two SVs that are unique to the fine silk strains relative to the coarse silk strains, which appeared to be an ideal candidate gene for fiber fineness. Thus we knocked out *BmChit β-GlcNAcase* by targeting its coding sequence to test the role in silk fineness determination to provide the first piece of functional evidence. We don't know whether the two SVs caused the differential expression of *BmChit β-GlcNAcase* right now. To confirm the effect of these two SVs (11.1 kb insertion and a 6.2 kb insertion) on gene expression, more experiments are required, such as knocked out these two SVs (11.1 kb insertion and a 6.2 kb insertion) separately in Chunfeng and Suxiu strains. But it could not be finished in a short time. In addition, we indeed agree that there might be some other genes affecting the silk fineness. Here, we would like to emphasis that *BmChit β-GlcNAcase* plays a role in silk fineness determination.

Regarding the description of knockout experiment, we thank the reviewer for noticing this mistake. In fact, CRISPR-cas9 mediated knockout of *BmChit β-GlcNAcase* produced coarser silk. We apologize for this mistake.

L299-325: This particular study of diapause might be exempted from the caveats of interpretation, because a previous QTL scan has determined that diapause differences can be attributed to genetic variation surrounding chromosome 11 (11-55.89cM). Thus, if the 747 bp deletion is the only variation

surrounding gene *BmTret1-like*, between the two phenotypes, the conclusion will be quite solid.
**Response:** The 747 bp deletion is the only variation surrounding gene *BmTret1-like* between *pnd* strain
(*pnd/pnd*) and wild type (+/+).

L326-358: I have no strong opinion on this study, as there are prior results indicating the function of *L* and
*Lc* locus, and the role of SVs is not overstated here.

**Response:** Thanks for your comment.

Minor Comments/Corrections:

L57. "We find that silkworm population harbors extremely variable genomes [...]". In what sense? Size
variation, structural variants, SNPs? Quantify it!

**Response:** High density of structure variants (one SV per 134 bp), Indel (one Indel per 49 bp) and SNPs
(one SNP per 11 bp) suggested that silkworm population harbors extremely variable genomes. This
sentence has been replaced with "We find that silkworm population harbors a high density of genomic
variants."

I would be intrigued to know HOW much larger the pangenome is than a typical sample genome. It seems
it might be quite high, with ~500 insertions, and an average of length of 10 kb, giving an average of an
extra 5 Mb. But I had the impression that my rough calculation could be an enormous underestimate.

**Response:** The pan-genome size (~9.6 Gb) is indeed much larger (~21 fold) than the typical sample
genome (~0.45 Gb). The pan-genome sequence consists of a linear reference genome (~0.45 Gb) and
all non-redundant insertions (~9.2 Gb). We have uploaded the pan-genome file to the China National
GeneBank DataBase (CNCBdb, <https://db.cngb.org>) under the accession number of CNP0002456.

L87. "We identified the SVs and their influencing genes underlying domestication and breeding processes
in silkworm". Consider rephrasing.

**Response:** We have revised this sentence as "We identified hundreds of SVs and genes potentially
underlying domestication and breeding of silkworm."

L97. It would be good at some point in the manuscript to have a brief description of what of each of these
categories (local strains, improved varieties, genetic stocks and wild silkworms) are and how they are
related to each other.

**Response:** The domesticated silkworms, including local strains and improved varieties as well as genetic
stocks, derived from wild silkworm ~5000 years ago. Local strains are breeding resources that were long
maintained in diverse geographic regions of the traditional silk producing countries and without further
cultivation. Improved varieties are further cultivated strains used for modern sericulture. Genetic stocks
are the natural mutants discovered during domestication and breeding improvement process, and artificial
mutants induced by chemical or physical treatment or genetic engineering. In the revised manuscript, we
have added above information in the "silkworm collection" of Methods of the revised manuscript. Please
see line 444-458.

L106-107. "[...] key determinants of the population structure of silkworms are artificial selection and [...]".

The distinction on PC1 between wild and domesticated populations is not necessarily explained by

artificial selection alone but by the domestication process (which includes drift and relaxed purifying
selection in neutral regions of the genome, not related to domestication traits).

**Response:** Thanks for your suggestion. We have revised this sentence as “Principal component analyses
(PCA) based on whole-genome SNPs of the 1,082 genomes showed that PC1 splits individuals into wild
and domestic groups, while PC2 further divides individuals into groups based on their geographic origin
in general.” in the revised manuscript.

L107. “The resources are divided [...]”. What does resources here mean? Probably there’s a better word.

**Response:** The resources represent all 1,082 strains. We have replaced “The resources” with “The 1,082
strains” in the revised manuscript.

L105-L110. These lines could be probably simplified to pass the main message which is that PC1 splits
individuals into wild and domestic groups, while PC2 further divides individuals into groups based on their
geographic origin.

**Response:** Thanks for your suggestion. L105-L110 have been revised as “Principal component analysis
(PCA) based on whole-genome SNPs of the 1,082 genomes showed that PC1 split individuals into wild
and domestic groups, while PC2 further divided individuals into groups based on their geographic origin
and improvement process in general (Extended data Fig 2a). Further, the result of phylogenetic analysis,
similar to the result of PCA, showed that the 1,082 strains are divided in wild group versus domestic
population which is further subdivided into the subclusters China-local, Europe-local, Tropical-local,
improved strains in China (CHN-I) and in Japan (JPN-I) (Fig. 1b,)” in the revised manuscript.

L110-112. I couldn’t understand what the authors meant in the second part of this sentence. Please clarify.

**Response:** We are sorry for that the sentence was not described clearly in the previous manuscript. Here,
we want to present that “The genetic stock strains are widely distributed in domesticated clades of the
phylogenetic tree (Fig. 1b), suggesting that these strains represent a wide genetic diversity. It could be
caused by that the genetic stock strains were generated by natural or artificial mutation during
domestication and breeding process of silkworm, and collected from worldwide.”. In the revised
manuscript, we have revised it as “Notably, the genetic stock strains are widely distributed in different
subclades of the domestic silkworm clade, (Fig. 1b), suggesting that these strains represent a wide genetic
diversity.”

L113. “The results generated four major subgroups”. Perhaps it would be better to rephrase to something
as “The results show the existence of four major subgroups”, since the groups already exist, and the
analyses only help describe them.

**Response:** Thanks, we have revised it based on your suggestion.

L130. “To reach an overview of genomic content in silkworms”. Perhaps replace “reach” by “give” or
“present”.

**Response:** We have replaced “reach” with “present” in the revised manuscript.

L135-138. It could be perhaps interesting to give a figure of how many of these assemblies are
chromosome level assemblies or have at least some entire chromosomes in a single scaffold.

**Response:** A silkworm genome contain 28 chromosomes. Thus, there are 15,260 (545*28) chromosomes

in the 545 long-read sequencing genomes. After *de novo* assembling, there are 971 chromosome-level
contigs in those genomes, about two chromosome-level contigs per genome (as shown in the Fig. 1
below). In the revised manuscript, we have revised the sentence as “*De novo* assemblies of these 545
genomes revealed an average genome size of 457.9 Mb, an average contig N50 size of 7.6 Mb (about
half the average length of a silkworm chromosome), and about two chromosome-level contigs per genome
(Fig. 2d).”.

Fig.1 Number of Chromosome-level contigs per genome

L138-139 – Here do the authors mean the percentage of “complete single copy BUSCOs” or does it
include also duplicated and fragmented BUSCOs? Also, is the mapping ratio the percentage of all
mapped reads or uniquely mapped reads?

**Response:** The percentage of BUSCO include complete single-copy, duplicated, and fragmented
BUSCOs. The mapping ratio is the percentage of all mapped reads. We added this information in Lines
571 and 573.

L159: “wildest” should be “widest”?

**Response:** Yes, it should be “widest”. We have corrected this mistake in the revised manuscript.

L172. Change from “Among which” to “Among these”.

**Response:** “Among which” has been replaced by “Among these genes” in the revised manuscript.

L180. Should also mention figure 3b.

**Response:** Yes, we have added figure 3b to this place in the revised manuscript.

L235-236. Perhaps it would be worth mentioning on what basis these candidates were identified (i.e. the
intersection of F_{ST} , Tajima’s D and XP-CLR).

**Response:** Thank you for your suggestion. In this study, we calculated population divergence index (F_{ST}),
neutrality tests (Tajima’s D), and the cross-population composite likelihood ratio test (XP-CLR) using SNP
markers. Those genes existed in overlapped genome regions of the top 1% F_{ST} , top 5% XP-CLR and
lowest 5% Tajima’s D_{local} (and Tajima’s $D_{local} < Tajima’s D_{wild}$) signatures were defined as domestication-
associated genes. We described the methods and threshold values of F_{ST} , Tajima’s D, and XP-CLR for

identifying candidates of domestication and breeding in Lines 538-548. After revised, the sentence is “We
calculated population divergence index (F_{ST}), neutrality tests (Tajima’s D), and the cross-population
composite likelihood ratio test (XP-CLR) using SNP markers. We defined the intersections of F_{ST} , Tajima’s
D and XP-CLR as selective sweep regions and identified 468 (2.8% of the whole-genome genes)
domestication-associated genes (Fig. 4a, Supplementary Table 8), containing 264 newly identified
domestication-associated genes compared with previous studies^{19, 36, 37}.” Please see Line 242-247.

Fig 1: CHN-I and JPN-I cannot be found on the spatial map. Does that mean they are only present in
laboratories?

**Response:** It is difficult to trace back the locations of these improved strains. Because they are practical
races that breed by using breeding techniques (traditional breeding and cross breeding, etc.) and are
preserved in various sericulture institutes. In general, they have two or more parent lines with different
genetic background (derived from different places). Therefore, we can not locate the places of improved
silkworms on the spatial map.

Also, I felt there should be a scale on the y axis of Fig. 1b -- it would be interesting to get some idea how
divergent the wild populations are from the domesticated populations. You could also use "Neighbor-
joining tree" instead of phylogenetic tree, since this is not really a phylogeny, but a distance-based
representation of strains within a species.

**Response:** Thank you for your comment and suggestion. In fact, the tree showed in Fig. 1b is a Neighbor-
joining tree. We have added the tree scale to Fig. 1b in the revised manuscript (Please see following Fig.
2 in this response).

Fig. 2 Phylogenetic tree of 1,082 genomes

Fig 3d: Many rings lack a y-axis (or description of the range of y-axis)

Response: We have added y-axis to fig. 3d in the revised manuscript (Please see following Fig. 3 in this
response).

**Fig. 3** Distribution map of genetic variations in 1,082 genomes

Fig 3a & 3c: Both figures have x-axis labeled as “Number of samples”, but if I understand correctly, 3a
means “each individual sample”, and 3c means “the number of included samples”.

**Response:** Yes, x-axis of fig 3a and 3c should be labeled as “each individual sample” and “the number of
included samples”. We have corrected it in the manuscript (Please see following Fig. 4 in this response).

**Fig. 4** Characterization of SVs in 545 silkworm genomes

**Response to Reviewer #2:**

GENERAL COMMENTS

Key Results

This manuscript reports construction of a pan-genome for the domesticated silkworm, *Bombyx mori*, and
its nearest wild ancestor, *B. mandarina*, starting with de novo next-generation sequences (NGS) of more
than 1000 silkworms from well-maintained, diverse collections of stocks used for practical breeding (so-
called “local” or genetically “improved” strains), and maintenance and analysis of genetic mutants (called
“genetic stocks” carrying documented morphological and biochemical mutations), and from wild
populations of *B. mandarina* collected in China. The authors conducted principle component analysis on
data for single nucleotide polymorphism (SNPs) from this large-scale NGS project to investigate
determinants of population structure which have a bearing on the history of silkworm domestication and
geographic origins of modern strains. From the phylogenetic tree for these data the authors then chose
545 representative strains to construct a high fidelity deep coverage pan-genome using nanopore long-
read sequencing. The report includes standard parameters to assess completeness of a genome
assembly (e.g., average read depth, length and total genome size, fraction of repeated sequences,
364 percent of well-conserved single copy orthologs or BUSCO, etc.), and to assess coverage and
365 completeness of a pan-genome, such as minimum number of long-read genomes needed to fully
represent the source populations (determined to be 100 in this case), its basic gene information content
(such as number of genes and gene families, assignment of sequences to “core” (shared by all samples),
softcore (shared by >90% but <100% samples), dispensable (shared by >1 but ≤ 90% samples) and
private (1% samples) categories, the properties and numbers of structural variants (SVs) and
transposable elements (TEs) relative to the published reference genome for silkworm and to pan-
genomes for *Drosophila melanogaster* and human, and so on. In their analysis and interpretation of these
and other data the authors consistently made the distinction between domesticated, genetically improved,
and wild silkworms, emphasizing the relevant impact of domestication and subsequent stock development
and maintenance, important subjects for silkworm scientists in the data.

Of broad interest is the authors’ treatment of the “Impact of SVs on genes,” in which they report the
proximity of indels, inserts, and so on in genes’ flanking and coding regions and introns (defined as
“potential expression regulatory regions” or PERRS), assign GO terms, and use RNA-seq to measure
expression in relatively small but apparently effective number (14) of silkworm strains with or without a
target SV. Interestingly, roughly 9% showed differential expression in at least one of 6 tissues used for
this study. Given how generalized these data sets are, it is encouraging that the authors were able to find
published evidence for predicted SV involvement in 9 experimentally characterized silkworm mutants
included in the pan-genome itself (Suppl. Table 6a). My guess is that a more extensive review of the
literature will find a similar association of SVs in most (all?) published silkworm mutations analyzed by
laborious traditional positional cloning methods. Being able to initiate a search for more subtle and likely
polygenic mutations using the fruits of the kind of analysis reported here should enable finding many more
SVs affecting expression of important genes in a kind of reverse functional analysis. The authors
demonstrated the potential strength of this approach by defining more than 400 “domestication-associated
genes” in a comparison of SV-associations between wild and domesticated pan-genomes, more than half

of which are newly described compared to previous studies. The authors took a similar approach to identify
potential “improvement-associated SVs” by comparing pan-genomes of improved Chinese (CHN-I) and
Japanese (JPN-I) to the corresponding local strains, again identifying many potential new targets for
further analysis. Readers will be able to peruse the extended tables associated with these studies to see
whether their favorite genes meet criteria as candidates for their own research.

The authors provide 6 concrete examples illustrating the success of this pan-genome approach for
functional genomics by newly identifying 2 genes affecting key sericultural traits (silk yield and fineness),
and 1 each affecting embryonic diapause (using a streamlined positional cloning approach on a well-
known homozygous diapause mutant), and a larval body marking. In all cases they confirmed the
identifications with targeted gene knockouts and in one case ectopic expression. Although the report of
these findings could probably stand on their own in a separate publication, presenting it here stands as a
strong, concrete illustration of the power of the new pan-genome for important applications in silkworm
biology, genetics, and functional genomics.

Validity and robustness of data and analytic approach

The report presents extensive data to support and validate the construction and composition of a silkworm
pan-genome, its properties, and potential applications. I do not have enough technical expertise in
bioinformatics or statistics to evaluate these aspects of it effectively so I will not comment further on them
except in general terms in other areas of this report.

Significance

This project addresses many major issues which are relevant and important for silkworm biology. As a
central theme the focus on the molecular genetic (genomic) basis of domestication, in which traits derived
from *B. mandarina* were ultimately fixed by selection, is estimated to have begun more than 5000 years
ago. The ongoing selection for traits favorable to sericulture and capture of spontaneous and induced
mutations which arose over the years have resulted in a highly diverse collection of genetic material which
has yet to be effectively mined for a heritage which is unique among insects. As noted in the report, some
of these characters were studied previously but with relatively little enlightenment regarding their
molecular basis and thus remain unexplained. Here the authors provide substantial evidence for the idea
that a pan-genomic approach can not only answer longstanding questions about the history of sericulture
but, equally important, uncover many more subtle and genetically complex traits which promise to open
up new areas for future study in silkworms and other insects.

Clarity, Context and Suggested improvements

For the most part I found the report to be relatively complete, clear, and well-written, with a minimum of
common misuses of fine points of English grammar (authors please see some detailed comments below
and written directly on the manuscript). A relatively minor exception is that, whereas the overall
manuscript’s organization of Introduction, Results, Methods, and Discussion is as expected, with mostly
necessary and appropriate information contained in those sections, I found some crossing of those
boundaries with information I think can be presented more effectively in a different section. This occurred

mostly in Methods where, after describing a procedure or protocol, the authors summarized (or repeated)
the findings and conclusions from that part of the project. Strictly speaking, I believe most of that kind of
information belongs in Results (not Methods) where it is essential for readers to know and understand the
nature and quality of the authors' findings. Further, not only may finding it reported under Methods
confound readers' expectations, it may also be missed by readers who (for various reasons) skim or skip
reading the Methods section.

**Response:** Dear Prof. Goldsmith, we thank you very much for your valuable time, detailed revision and
constructive comments on our manuscript. According to your suggestions in the comment file and in the
manuscript text file with track changes, we have thoroughly revised the manuscript. The revised parts of
the manuscript have been highlighted in yellow. We have done a point-by-point reply for your suggestions
and comments below. In addition,

Additional General and Specific Comments about content are listed separately below under each section
of the manuscript.

General Comments About Writing And Expression

Although the manuscript as a whole reads well (as noted above), I urge the authors to be consistent in
their use of verb tense. In my understanding the general policy for scientific journals is to report new
results in a manuscript in the past tense (e.g., the indicating authors "did" something and "found X, Y, Z").
In contrast, published results are reported in the present tense as "true" or existing "facts." And hypotheses
and conclusions are (perhaps confusingly?) written in the present tense as being newly reported. Here
the authors usually report their findings in the present tense, which is not consistent with these "rules" but
okay with me if also okay with the journal. However, in a few places the authors use past tense to report
their new results or observations. I have marked these cases in the manuscript when I noticed the
difference but did not try to do this consistently and leave changes for overall consistency to the authors
(or copy editors?).

**Response:** Thanks, based on your suggestion, we have checked and revised the verb tenses throughout
the manuscript.

In several places I recommend deleting "respectively" as unnecessary where the word order of items in
successive lists within a single sentence is obvious. Or (at least in one case) there are no successive lists.
Again, specific examples of this are marked on the manuscript and/or noted below.

**Response:** We have deleted all unnecessary "respectively" in the revised manuscript.

Suggestions for Content and Specific Line-By-Line Comments by Manuscript Section

Introduction

No specific (line) comments here. I found this section to be relevant and concise. Although the authors
might consider adding a somewhat more general article (a summary or review) on the current status of
pan-genomics in addition to those cited, such as Golicz et al. (2020) Trends in Genetics
<https://doi.org/10.1016/j.tig.2019.11.006>. And possibly consider citing a recent bioRxiv prepublication
reporting a 3-species pan-genome for members of the *Heliconius* butterfly clade designed to examine the

evolution of chromatin accessibility (Ruggieri et al.,
<https://www.biorxiv.org/content/10.1101/2022.04.14.488334v1>). The appearance of this and the present
publication suggests a pan-genome approach will soon be used more often for moths and butterflies,
especially given a rapid increase in the number of reference genomes for these clades (e.g., see Ellis et
al., <https://doi.org/10.1093/gigascience/giab041> and Darwin Tree of Life Project, among other possible
sources).

**Response:** Thanks for your suggestion. The review article (Golicz et al., 2020, Trends in Genetics) and
the bioRxiv prepublication of butterfly have been cited in the introduction section of the revised manuscript.
Please see reference 6 and 14 in the revised manuscript.

RESULTS

General comments

I found most of the content of Results to be of interest and reasonable, with one exception, notably, the
authors' treatment of multigene families. At first I found somewhat startling their assertions of having found
so many multigene families – indeed, a great many more than in the last published reference genome for
silkworm, which is reasonably complete. The authors identify sequences defined here as members of
multigene families by standard bioinformatic criteria and organize and classify them in what I assume are
standard ways used in genomics and pan-genomics such as annotation using various resources (NR, GO,
KEGG, and KOG terms) with resulting amino acid sequences, evidence for expression via RNA-seq
(carried out by the authors) reported as FPKM values, and assignment to core, softcore, and dispensable
categories based on the degree of sequence conservation among the strains comprising the pan-genome.
Inspection of supplementary Table 4a which contains these data revealed a wide assortment of types of
sequences, including transposable elements, viruses, reverse transcriptases, untranslated RNAs, etc. etc.,
together with genes coding for well-studied kinds and classes of structural proteins. Although this
somewhat simplified approach to characterizing and classifying multigene families may be reasonable in
terms of “pure” bioinformatics where annotations rely on somewhat arbitrary criteria for cataloguing and
grouping sequences, it leaves out an important consideration of gene function, which results in vastly
different structural constraints and consequently evolutionary properties, depending on whether a
sequence encodes a viral remnant, a transposable element, a reverse transcriptase, a noncoding RNA,
or a structural protein, to name just a few examples. The resulting number of family members, degree of
sequence identity, and inferred evolution of these different types of “genes” will be vastly different, and,
perhaps, should not be considered and reported using the same general criteria.

A specific example of my problem with this pooling of data for all types of “multigene families” is the result
of a search for “chorion,” a multigene family for a class of structural proteins with which I am familiar,
where I found 34 sequences under the column of “NR” (NCBI annotation). This corresponds well to the
reported number of chorion genes (and proteins). However, here 27 of them are marked “y” for “newly
identified,” and only 7 are listed as “n” or not newly identified. Of the so-called “newly identified” chorion
proteins, 4 are listed as “softcore” or moderately conserved, and the remainder as “dispensable” (which
should be spelled “dispensable”), meaning not very well conserved; whereas roughly half (3) of the so-
called previously identified chorion proteins are listed as “softcore” and half (4) as “dispensable.” The
assignments to these categories and their inferred evolutionary histories simply don't jibe with the well-

established number, family member distribution and conservation of chorion proteins which are well-
 characterized at protein and DNA levels and have been examined in depth in two evolutionarily diverged
 silkworm strains, the genome reference strain Daizo and the European strain 703. This misdirection could
 simply be a result of the over-general way the gene-finding algorithms assign identity to families with
 structural members which are diverse at a micro-level but, nevertheless, maintain overlapping or even
 identical functions, which indicates they are basically well-conserved. In other words, it is unlikely that the
 27 labeled here as “newly identified” are actually “new;” it’s just that their sequences have diverged enough
 among each other (and from the original reference strain) to be defined as “new” by the limited
 bioinformatic criteria used here. Although I did not investigate this issue further, I am certain readers will
 have similar reservations about the treatment of other well-characterized multigene families which encode
 structural proteins that have been important research subjects in silkworm and other insects, such as
 cuticle proteins, detoxification enzymes, neural peptides, and so on. These observations suggest to me
 the authors should consider adding to the Results (perhaps in supplementary Table 4 or in another
 supplementary table) some sub-grouping of gene family members into at least rough functional categories.
 And add to the Discussion some information (and perhaps caveats) about the nature and identities of
 sequences listed as “multigene families” in this report.

**Response:** Thanks for your suggestion. In the previous manuscript, we define gene families based on
 similarity of gene sequences. We are very sorry that we mistakenly conflated gene families and
 orthologous group (Orthogroup) in the previous manuscript. Each row of Supplementary Table 4 in the
 previous manuscript represented an Orthogroup rather than a gene family. I have changed “gene family”
 to “Orthogroup” in the revised manuscript. Orthogroups are identified using the following strategy.
 OrthoFinder was used to first cluster genes based on sequence similarity, and then classify the
 homologous genes into different Orthogroups based on the gene tree. This method will generate an
 Orthogroup that may have different numbers of genes in different samples. For example, five genes of the
 Daiao strain are classified into the Orthogroup OG0000241 that is annotated (NR, NCBI annotation) as
 chorion class CB protein M5H4-like (Fig. 5, line marked with blue background). We did find 27 chorion
 protein Orthogroups (Fig. 5, Orthogroup marked with green background) that absent in the reference
 genome (Daiao strain). We thus named these 27 Orthogroups as newly identified genes.

Gene count of each orthogroup in each strain (partial)

Orthogroup	Daiao	BomL86	BomP200	BomM348	BomP54	BomL197	NR_annotation
OG000011	0	26	25	24	11	10	XP_004924960.1 chorion class A protein L11-like [Bombyx mori]
OG000059	0	22	20	13	4	2	BAS21447.1 chorion class B [Bombyx mori]
OG000174	0	7	4	5	4	6	XP_028028831.1 chorion class high-cysteine HCB protein 13-like [Bombyx mandarina]
OG000233	2	7	7	6	4	1	AAA27833.1 early chorion protein precursor, partial [Bombyx mori]
OG000241	5	5	5	6	5	4	XP_004933236.1 chorion class CB protein M5H4-like [Bombyx mori]
OG000476	3	3	3	1	4	0	NP_001112378.1 chorion class CA protein ERA.3 precursor [Bombyx mori]
OG000546	2	4	4	3	3	0	XP_028028815.1 chorion class B protein ERB4 [Bombyx mandarina]
OG000664	0	3	2	2	1	3	XP_028042482.1 chorion class B protein M3A5-like [Bombyx mandarina]
OG001536	0	2	2	2	1	0	BAS21376.1 chorion early B [Bombyx mori]
OG002090	0	0	0	1	1	0	pir B21761 high cysteine chorion B 12 protein precursor - silkworm
OG010449	1	1	1	1	1	1	XP_004924968.2 chorion class CA protein ERA.2-like isoform X1 [Bombyx mori]
OG010795	1	1	1	1	1	1	XP_028041163.1 chorion class CB protein M5H4-like [Bombyx mandarina]
OG011458	0	1	1	1	0	0	XP_028028767.1 chorion class B protein ERB4-like [Bombyx mandarina]
OG011589	1	1	1	1	1	0	XP_004933235.1 chorion class B protein PC10 [Bombyx mori]
OG011746	0	1	1	1	1	1	XP_004933236.1 chorion class CB protein M5H4-like [Bombyx mori]
OG011833	1	1	1	1	1	0	XP_028028843.1 chorion class CB protein M5H4-like [Bombyx mandarina]
OG011857	1	1	1	1	0	1	XP_028028648.1 chorion class A proteins Ld9-like [Bombyx mandarina]
OG012117	1	1	1	0	0	1	XP_028028845.1 chorion class B protein ERB4-like [Bombyx mandarina]
OG012156	1	1	1	1	0	0	XP_012546877.2 chorion class B protein M2807-like [Bombyx mori]
OG012204	0	1	0	1	0	1	XP_004927417.1 chorion class A protein L11 [Bombyx mori]
OG012255	0	1	1	1	0	0	XP_028042477.1 chorion class A protein L12-like [Bombyx mandarina]
OG012296	0	1	1	0	1	0	XP_028028836.1 chorion class B protein ERB4-like [Bombyx mandarina]
OG012394	0	1	1	1	1	0	XP_004933242.1 chorion class CA protein ERA.4-like [Bombyx mori]
OG012726	0	1	0	0	4	1	XP_028028702.1 chorion class B protein M2410-like [Bombyx mandarina]
OG012851	0	0	1	1	0	0	XP_021202863.1 chorion class B protein L12-like [Bombyx mori]
OG012860	0	0	1	0	0	0	BAS21367.1 chorion early B [Bombyx mori]
OG013000	0	0	2	1	0	0	BAS21375.1 chorion early B [Bombyx mori]
OG013268	0	0	1	0	0	0	XP_004925015.1 chorion class B protein M2410-like, partial [Bombyx mori]
OG013662	0	0	1	0	1	0	XP_004924965.1 chorion class A protein L12-like [Bombyx mori]
OG013669	0	0	0	0	1	1	BAS21377.1 chorion class B [Bombyx mori]
OG013729	0	0	1	0	0	1	XP_012544945.1 chorion class CA protein ERA.1 [Bombyx mori]
OG014326	0	1	1	0	0	0	XP_028041159.1 chorion class B protein L11-like [Bombyx mandarina]
OG014742	0	0	0	0	1	0	XP_028028803.1 chorion protein ERB.1-like [Bombyx mandarina]
OG014933	0	0	0	0	1	1	XP_004927434.1 chorion class B protein L12-like [Bombyx mori]
OG015254	0	0	0	0	0	1	XP_028028639.1 chorion class A protein M2774-like [Bombyx mandarina]
OG015326	0	0	0	1	1	0	P08830.2 RefName: Full=Chorion class CB protein M5H4; Flags: Precursor
OG015921	0	0	0	1	0	0	XP_021208710.1 chorion class high-cysteine HCB protein 12-like [Bombyx mori]
OG016805	0	1	0	0	0	0	BAS21446.1 chorion class A [Bombyx mori]

Fig. 5 Gene count of each orthogroup of each strain (partial).

In addition, we also found that the number of chorion genes varies greatly among different genomes (Fig. 6 in the response). For example, there are 19 chorion genes in the Dazao genome, while the number of chorion genes in BomM148 strain reaches 104 (Fig. 6, marked with red arrow). We guess the diversity of chorion gene numbers may be related to the rich phenotypic diversity of silkworm eggs, but this still needs to be supported by more experimental evidence in the future.

Fig.6 The number of chorion genes in each strain

Finally, we strongly agree with you that it is indeed difficult to accurately classify gene families based on sequence similarity or bioinformatic criteria alone. We tried to classify gene families based on gene function, but we found it very difficult to classify all gene families at the genome-wide level. We first tried to classify genes based on the annotation results of "NR". For instance, we searched with "chorion" as the keyword, and found that there are 40 Orthogroups whose NR annotation column contains "chorion", but there are two Orthogroups annotated as "chorion peroxidase" instead of chorion gene (marked with yellow in Fig. 7). We also investigated other gene families, such as zinc finger protein and P450 gene families, and we encountered similar problem to the analysis of chorion gene family. We further tried to classify genes according to the functional annotation results of "GO", but we found that different genes in the same gene family will appear different GO terms. For example, among the 38 chorion Orthogroups of silkworm, there are 32 Orthogroups with GO numbers as "GO:0042600; GO:0005213; GO:0007304; GO:0007275", and 6 Orthogroups are marked as "NA"(marked with green in Fig. 7). It is difficult to obtain accurate results when classifying gene families based on GO information. We have not yet found a good way to accurately classify all multigene families according to gene function, which may be the common problem encountered by most of the current genome research. Therefore, we are very sorry that we cannot provide results for all multigene families in the revised manuscript. In contrast, it is easier to identify a specific gene family in the whole genome, which usually requires separately analysis combining full sequence similarity and conserved domains. We have submitted all gene information and sequences for each strain to a public database, which will allow researchers to analyze multigene families of their interest.

Supplementary Table 4. Information of pan-gene (partial)			
Order	GeneFamilyID	NR_annotation	GO_annotation
OG0000011	XP_004924960.1 chorion class A protein L11-like [Bombyx mori]	NA
OG0000059	BAS21447.1 chorion class B [Bombyx mori]	GO:0042600, GO:0005213, GO:0007304, GO:0007275
OG0000174	XP_028028831.1 chorion class high-cysteine HCB protein 13-like [Bombyx mandarina]	GO:0042600, GO:0005213, GO:0007304, GO:0007275
OG0000233	AAA27833.1 early chorion protein precursor, partial [Bombyx mori]	GO:0042600, GO:0005213, GO:0007304, GO:0007275
OG0000241	XP_004933236.1 chorion class CB protein M5H4-like [Bombyx mori]	GO:0042600, GO:0005213, GO:0007304, GO:0007275
OG0000476	NP_001112378.1 chorion class CA protein ERA 3 precursor [Bombyx mori]	GO:0042600, GO:0005213, GO:0007304, GO:0007275
OG0000546	XP_028028815.1 chorion class B protein ERB4 [Bombyx mandarina]	GO:0042600, GO:0005213, GO:0007304, GO:0007275
OG0000664	XP_028042482.1 chorion class B protein M3A5-like [Bombyx mandarina]	GO:0042600, GO:0005213, GO:0007304, GO:0007275
OG0001536	BAS21376.1 chorion early B [Bombyx mori]	GO:0042600, GO:0005213, GO:0007304, GO:0007275
OG0002090	pirj B21761 high cysteine chorion B 12 protein precursor - silkworm	GO:0042600, GO:0005213, GO:0007304, GO:0007275
OG0002821	BBD52153.1 chorion peroxidase [Bombyx mori]	GO:0020037, GO:0004601, GO:0048477, GO:0006979
OG0010449	XP_004924968.2 chorion class CA protein ERA 2-like isoform X1 [Bombyx mori]	NA
OG0010795	XP_028041163.1 chorion class CB protein M5H4-like [Bombyx mandarina]	GO:0042600, GO:0005213, GO:0007304, GO:0007275
OG0011458	XP_028028767.1 chorion class B protein ERB4-like [Bombyx mandarina]	GO:0042600, GO:0005213, GO:0007304, GO:0007275
OG0011589	XP_004933235.1 chorion class B protein PC10 [Bombyx mori]	GO:0042600, GO:0005213, GO:0007304, GO:0007275
OG0011746	XP_004933236.1 chorion class CB protein M5H4-like [Bombyx mori]	GO:0042600, GO:0005213, GO:0007304, GO:0007275
OG0011833	XP_028028843.1 chorion class CB protein M5H4-like [Bombyx mandarina]	GO:0042600, GO:0005213, GO:0007304, GO:0007275
OG0011857	XP_028028648.1 chorion class A proteins Ld9-like [Bombyx mandarina]	NA
OG0012117	XP_028028845.1 chorion class B protein ERB4-like [Bombyx mandarina]	GO:0042600, GO:0005213, GO:0007304, GO:0007275
OG0012156	XP_012546877.2 chorion class B protein M2807-like [Bombyx mori]	GO:0042600, GO:0005213, GO:0007304, GO:0007275
OG0012204	XP_004927417.1 chorion class A protein L11 [Bombyx mori]	NA
OG0012255	XP_028042477.1 chorion class A protein L12-like [Bombyx mandarina]	GO:0042600, GO:0005213, GO:0007304, GO:0007275
OG0012296	XP_028028836.1 chorion class B protein ERB4-like [Bombyx mandarina]	GO:0042600, GO:0005213, GO:0007304, GO:0007275
OG0012394	XP_004933242.1 chorion class CA protein ERA.4-like [Bombyx mori]	NA
OG0012726	XP_028028702.1 chorion class B protein M2410-like [Bombyx mandarina]	GO:0042600, GO:0005213, GO:0007304, GO:0007275
OG0012851	XP_021202863.1 chorion class B protein L12-like [Bombyx mori]	GO:0042600, GO:0005213, GO:0007304, GO:0007275
OG0012860	BAS21367.1 chorion early B [Bombyx mori]	GO:0042600, GO:0005213, GO:0007304, GO:0007275
OG0013000	BAS21375.1 chorion early B [Bombyx mori]	GO:0042600, GO:0005213, GO:0007304, GO:0007275
OG0013268	XP_004925015.1 chorion class B protein M2410-like, partial [Bombyx mori]	GO:0042600, GO:0005213, GO:0007304, GO:0007275
OG0013662	XP_004924965.1 chorion class A protein L12-like [Bombyx mori]	GO:0042600, GO:0005213, GO:0007304, GO:0007275
OG0013669	BAS21377.1 chorion class B [Bombyx mori]	GO:0042600, GO:0005213, GO:0007304, GO:0007275
OG0013729	XP_012544845.1 chorion class CA protein ERA.1 [Bombyx mori]	GO:0042600, GO:0005213, GO:0007304, GO:0007275
OG0014326	XP_028041159.1 chorion class B protein L11-like [Bombyx mandarina]	NA
OG0014742	XP_028028803.1 chorion protein ERB.1-like [Bombyx mandarina]	GO:0042600, GO:0005213, GO:0007304, GO:0007275
OG0014933	XP_004927434.1 chorion class B protein L12-like [Bombyx mori]	GO:0042600, GO:0005213, GO:0007304, GO:0007275
OG0015254	XP_028028639.1 chorion class A protein M2774-like [Bombyx mandarina]	GO:0042600, GO:0005213, GO:0007304, GO:0007275
OG0015326	P08830 2 RecName: Full=Chorion class CB protein M5H4, Flags: Precursor	GO:0042600, GO:0005213, GO:0007304, GO:0007275
OG0015921	XP_021208710.1 chorion class high-cysteine HCB protein 12-like [Bombyx mori]	GO:0042600, GO:0005213, GO:0007304, GO:0007275
OG0016805	BAS21446.1 chorion class A [Bombyx mori]	GO:0042600, GO:0005213, GO:0007304, GO:0007275
OG0018447	XP_028032358.1 chorion peroxidase-like [Bombyx mandarina]	GO:0020037, GO:0004601, GO:0006979

Fig. 7 Information of each orthogroup

**Specific Comments**

Lines 248-250: misuse of “respectively” (please see comments above; “respectively” is not used when the
word order is clear)

We identified 126 (CHN-I)

and 116 (JPN-I) improvement-associated regions (IARs) containing 106 and 92

improvement-associated genes (IAGs), respectively (Fig. 4c, Supplementary Table 8).

**Response:** We have deleted unnecessary “respectively” in the revised manuscript.

Lines 228-231 leave me with some confusion. 1560/2396 = 65% - so I do not understand where the value
of 9.2% comes from

228 Finally, 1,560 genes

in 2,396 SV-gene pairs (9.2%) were found to be differentially expressed (FDR <

0.001) in at least one tissue between strains with and without corresponding SV

(Extended Data Fig. 4h), indicating a large impact of SVs on gene expression.

**Response:** The ratio (9.2%) was calculated by 2,396/26,188 SV-gene pairs. A total of 26,188 SV-gene
pairs were generated from 14 RNA-seq samples. Among these 26,188 pairs, 2,396 pairs contain 1,560
genes that show differentially expressed (FDR < 0.001) in at least one tissue between strains with and
without corresponding SV. Here, there are a total of 1,560 genes in the 2,396 SV-gene pairs, because
multiple SVs may be located near or within the same gene region. In the revised manuscript, we have
revised this sentence as “Among these pairs, 2,396 SV-gene pairs (9.2%) contained a total of 1,560 genes
that showed differentially expressed (FDR < 0.001) in at least one tissue between strains with and without
corresponding SV (Extended Data Fig. 4h)”.

Lines 235-237: Authors, please indicate here how you define or describe the term “domestication-
associated” genes (DAGs). I believe you did this later in the manuscript but it would be better to do it the
first time the term appears.

desirable traits in silkworm. We identified 468 (2.8% of the whole-genome genes)

domestication-associated genes (DAGs) (Fig. 4a, Supplementary Table 8), containing

264 newly identified DAGs compared with previous studies^{19, 36, 37}.

**Response:** Thank you for your suggestion. In this study, we calculated population divergence index (F_{ST}),
neutrality tests (Tajima's D), and the cross-population composite likelihood ratio test (XP-CLR) using SNP
markers. Those genes existed in overlapped genome regions of the top 1% F_{ST} , top 5% XP-CLR and
lowest 5% Tajima's D_{local} (and Tajima's $D_{local} < Tajima's D_{wild}$) signatures were defined as domestication-
associated genes. We described the methods and threshold values of F_{ST} , Tajima's D, and XP-CLR for
identifying candidates of domestication and breeding in Lines 538-548. After revised, the sentence is “We
calculated population divergence index (F_{ST}), neutrality tests (Tajima's D), and the cross-population
composite likelihood ratio test (XP-CLR) using SNP markers. We defined the intersections of F_{ST} , Tajima's
D and XP-CLR as selective sweep regions and identified 468 (2.8% of the whole-genome genes)
domestication-associated genes (Fig. 4a, Supplementary Table 8), containing 264 newly identified
domestication-associated genes compared with previous studies^{19, 36, 37}.” Please see Line 242-247;

Lines 284-286: Please define or explain SX, CF, XF, and QB which appear here for the first time in the
MS. It is possible the authors mean to indicate SX and CF are associated with fine silk, and XF and QB
are associated with coarse silk. However, the way parentheses () are used here confuses rather than
clarifies the situation, at least for me.

Here we performed RNA-seq of the silk press in fine silk (SX

(BomP174), CF (BomP79)) and coarse silk (XF (BomP154), QB (BomP31)) strains

(Fig. 5e, Extended Data Fig. 7a).

**Response:** We have revised this sentence as “Here we performed RNA-seq of the silk press in four
strains including two fine silk strains (Suxiu, Chunfeng) and two coarse silk strains (Xiafang, Qiubai) (Fig.
5e, Extended Data Fig. 7a).”.

Lines 356-358 I was confused at first by the authors' use of “mapping cloning.” I believe a better term is
“map-based” cloning.

These results reveal that large and complex SVs in L alleles, which cannot be obtained

by mapping cloning, affect the expression pattern of Wnt1 and result in twin-spot

markings.

**Response:** Thanks, we have replaced “mapping cloning” with “map-based cloning” in the revised
manuscript.

DISCUSSION

General Comments

In addition to the suggestion above under Results and one below under Methods, Perhaps it would be
helpful to readers to suggest other areas for future study which are associated with silkworm

domestication and have been relatively refractory to a “classic” trait-based genetic mapping/sequencing
approach. For example, differences in *B. mori* behavior relative to *B. mandarina*, such as larval ability to
withstand crowding and handling, relatively docile feeding (lacking a strong drive for finding food), and
loss of flight ability. Judicious choice of these could lead to new insights in other Lepidoptera or even in
other kinds of insects.

**Response:** Thanks for your suggestion. We have added these points in the “Discussion section” of the
revised manuscript. Please see Line 398-402 “The analysis of the functions of these genes will reveal the
genetic basis of artificial selection and provide improvement targets, as well as promote our understanding
of differences in *B.mori* behavior relative to *B. mandarina*, such as larval ability to withstand crowding and
handling, relatively docile feeding (lacking a strong drive for finding food), and loss of flight ability.”.

Specific Comments

Lines 378-380: I question the breadth of this statement with reference to its use of “various species.” It
just seems a little too vague and general. Specifically, I question the extent to which findings in silkworms
will help in the understanding the domestication of any other ANIMAL species (plants less unlikely) except
perhaps for insects, but probably not mammals or avians, which are the main species I believe we can
say have been “domesticated”. And given that (in my understanding) no insect except silkworm is truly
domesticated, i.e., having been so strongly selected for traits of value to humans that it is, effectively,
significantly different from its nearest wild ancestor (and therefore merits being called a separate species)
and fully dependent on us for survival, this statement would be more convincing if focussed (at least) on
insects, or even, possibly, only on Lepidoptera and, perhaps, honeybees.

The analysis of the functions of these DAGs and IAGs will reveal the genetic basis of
artificial selection and provide improvement targets, as well as help the understanding
of the common genetic mechanisms underlying domestication of various species.

**Response:** Thanks for your suggestion. As described above, we have revised this sentence as “The
analysis of the functions of these genes will reveal the genetic basis of artificial selection and provide
improvement targets, as well as promote our understanding of differences in *B.mori* behavior relative to
*B. mandarina*, such as larval ability to withstand crowding and handling, relatively docile feeding (lacking
a strong drive for finding food), and loss of flight ability.”.

Lines 384-388: I find this statement to be clear enough but it is unnecessarily repetitious (stating silkworm
economic traits twice in two short sentences). Further, it could be expanded to suggest broader
applications. I suggest changing it to something like the following (perhaps leaving out the underlined
phrases as overstated and unnecessary since they refer to a continuing theme of the report):

“Furthermore, our use of a large-scale pan-genome to decipher two genes that control important economic
traits in silkworms may also be used to reveal genetic mechanisms and traits associated with the survival
of wild populations and evolution of new species under strong natural selection by human and non-human
factors.”

Furthermore, we

deciphered two genes (BmE2F1 and BmChit β -GlcNAcase) that control important

economic traits in silkworms relevant to silk yield and fineness using large-scale pan-

genome. These findings have significance for improving economic traits of silkworm

varieties.

**Response:** Thanks for your suggestion. In the revised manuscript, we have revised this sentence as
"Furthermore, our use of a large-scale pan-genome to decipher two genes that control important economic
traits in silkworms may also be used to reveal genetic mechanisms and traits associated with the survival
of wild populations and evolution of new species under strong natural selection by human and non-human
factors". Please see line 407-410.

Methods

Lines 429-432: Although this statement contains basic information about the experimental design in terms
of numbers of samples of various categories the authors used in this research, it also contains a specific
comparison with previous studies which I suggest is more appropriately reported in a Discussion section
than in Methods.

of their geographic distributions in China. We have a larger sample size and a wider
geographic distribution of sample set compared with previous publications that
contained 40 (11 wild silkworms and 29 domestic silkworms) and 144 (seven wild
silkworms and 137 domestic silkworms) strains in 200936 and 201819.

**Response:** Thank you for raising these points. This information has been moved to the discussion section
of the revised manuscript. Please see line 377-381.

Line 469: Authors: Please clarify or explain what you mean by the term "regular." Perhaps replace with a
more technical descriptive term.

469 kb de novo regular library of each sample was used to sequence on PromethION

**Response:** The regular library means a standard sequencing Library. The "de novo regular library" has
been replaced with "DNA library" in the revised manuscript.

Lines 531-536: These lines describe new data from the study. For reasons noted above I believe this
information belongs in the Results section and should be removed from Methods. I have a similar concern
about information reported in lines 540-543, 576-581, 609-612, 616-625, and 692-695 which are
underlined in the text but not extracted here.

For domestication, 468 genes were identified as potential

domestication-associated genes (DAGs). Comparing to previous studies¹⁹, 36, 37, we
newly identified 264 DAGs in our extended panels of wild and domestic silkworms.

For improvement, we identified 189 improvement-associated genes (IAGs) containing

nine genes shared by CHN-I and JPN-I. 185 of those genes are newly identified

compared with IAGs in the study of Xiang et al¹⁹.

**Response:** Thanks for your suggestion. We have moved these contents to the results section in the
revised manuscript. For instance, L531-536, L576-581, L609-612, and L692-695 of the previous
manuscript have been moved to L246-263, L138-142, L152-153, and L269-271 of the revised manuscript.

In addition, we have deleted L540-543 and L616-625 in the revised manuscript. Because both results are
not important and difficult to integrate into the proper location of the "Results section" in the revised
manuscript. For example, the Line 540-543 "The average estimates of genome size, the rate of
heterozygosity, and repetitive elements ratio for those 545 genomes are 449 Mb, 0.53%, and 51%,
respectively (Supplementary Table 3)." are the results of genome survey predicted by genomeScope v1.0

using NGS data. It gives a preliminary understanding of the genomic characteristics before assembly, but
not the real characters of these genomes. So we deleted this information in the revised version.

Line 703: Shouldn't the authors list here the helper plasmid used in the CRISPR-cas9 procedure?

The transgenic vector piggyBac [3×P3-EGFP, Fib-H-BmE2F1-SV40] was

constructed to over express BmE2F1 gene in silk gland. The vector was injected with

the helper plasmid into newly laid eggs by microinjection.

**Response:** The helper plasmid is pHA3PIG containing the piggyBac transposase sequence and the *B.*
*mori actin 3* promoter (Tamura et al., 2000, Nat Biotechnol, 18:81-84). We have added this information in
the revised. Please see line 692-693.

(signed) Marian R. Goldsmith

Reviewers' Comments:

Reviewer #1:

Remarks to the Author:

We have now read the revised manuscript and we are satisfied that the authors have adequately addressed almost all issues identified by the reviewers, and we now look forward to seeing the publication of the revised manuscript, with the very minor recommendations for further clarification below.

We found it helpful to have changes in the manuscript marked in yellow. However, unlike the first version of the manuscript, it seemed to us that the new version of the manuscript had significant grammatical errors, especially in these sections of text marked as revised. Please check the English more carefully before submitting the final manuscript! We are not sure how much editorial help you might get in this respect from Nature Communications staff, but it is not acceptable English at present. We highlight some of the types of errors below. However, please note that it is not our job as reviewers to be exhaustive, and there are other examples of similar kinds of grammatical errors that we do not highlight.

We have not checked that all the genomes and the read data are actually available where the authors say they are. Can you check that all the assemblies and read data are actually present in the databases under the reference numbers given? We looked cursorily in the CNGdb but the database drew a blank when we searched for Project: CNP0002456, for example.

Suggested changes, mainly grammatical:

I. 56: We suggest adding articles, such as here: "...the entire genomic content in THE silkworm. We found that THE silkworm population..."

I. 98: We suggest: "... depth of ~65x PER SAMPLE were..."

I. 113: "... represent a wide genetic diversity." Still not entirely clear what you mean. Presumably you mean that these samples cover broadly across the diversity of domesticated silkworm and its wild progenitor species.

I. 143-144: We think you should say something like "... were 98-99% on average, INCLUDING SINGLE COPY, DUPLICATED AS WELL AS FRAGMENTED GENES (Fig. 2f, ...)". You do mention this in the methods, but it needs to be stated in the results.

I. 233: "... the expression of gene..." -- either "we next investigated GENE EXPRESSION ..." or "EXPRESSION OF GENES ..."

II. 263-4: "Compared with A previous study..."

I. 264: "... shared ONLY around 3% ..."

I. 266-7: "... the genetic bases of silkworm heterosis..." This seems unclear, unless these changes largely deleterious as homozygotes?

I. 395: " ... for survival. HOWEVER, only a few ..." It seems best to break a sentence here.

I. 449: "... and without further SELECTIVE BREEDING..." -- "cultivation" seems wrong here, since it implies tilling earth and rearing plants rather than rearing insects.

I. 452: "... further cultivated..." seems wrong here as well. Do you mean HIGHLY SELECTIVELY BRED strains?

I. 456: "... represent THE ancestor (BOMBYX MANDARINA) of THE domesticated silkworm."

I. 574: "... (included complete ...". We think the authors mean "(INCLUDING complete ...". There are a number of other cases where "included" is used for "including" elsewhere that should also be corrected.

Reviewer #2:

Remarks to the Author:

I appreciate the authors' careful attention to the points I made in my initial review regarding organization and English expression and have only a few further substantive comments below, plus some additional minor suggestions for improving English usage which I noted on the attached annotated manuscript pdf rather than excerpting and describe them here. I especially appreciate the authors' change in reporting as "orthogroups" what they originally termed "multigene families." The former seems better suited to analysis and understanding of the relatively diverse and complex data they have developed for such a large and comprehensive pan-genome.

Without wanting to belabor their follow up analysis of "multigene families" too much (which I understand is being submitted privately by way of explanation and not for publication with the actual manuscript), although I appreciate the authors' re-worked data analysis of chorion genes shown in Response Figures 5, 6, and 7, and I'm not surprised by their finding that chorion protein gene numbers varied significantly among the strains they examined, still, the number of members in each of the major "families" they presented here for Dazao do not seem to be as well correlated as I would expect with the published data on chorion genes in this strain which was based on detailed manual annotation of BAC clones and analysis of expression patterns (e.g., please see Chen et al. (2015) SCIENTIFIC DATA | 2:150062 | DOI: 10.1038/sdata.2015.62). Perhaps this simply illustrates some pitfalls in using global bioinformatic or algorithmic tools rather than detailed manual annotation for accurate descriptions of these kinds of complex protein families which not only have overlapping functions but also a potential for relatively rapid evolution.

Reviewer #3:

None

Reviewer #4:

None

Response to Reviewer #1

Reviewer #1(Remarks to the Author):

We have now read the revised manuscript and we are satisfied that the authors have adequately addressed almost all issues identified by the reviewers, and we now look forward to seeing the publication of the revised manuscript, with the very minor recommendations for further clarification below.

We found it helpful to have changes in the manuscript marked in yellow. However, unlike the first version of the manuscript, it seemed to us that the new version of the manuscript had significant grammatical errors, especially in these sections of text marked as revised. Please check the English more carefully before submitting the final manuscript! We are not sure how much editorial help you might get in this respect from Nature Communications staff, but it is not acceptable English at present. We highlight some of the types of errors below. However, please note that it is not our job as reviewers to be exhaustive, and there are other examples of similar kinds of grammatical errors that we do not highlight.

Response: Thank you for raising this point. We have carefully checked the full text and corrected all grammatical errors in the revised manuscript.

We have not checked that all the genomes and the read data are actually available where the authors say they are. Can you check that all the assemblies and read data are actually present in the databases under the reference numbers given? We looked cursorily in the CNGdb but the database drew a blank when we searched for Project: CNP0002456, for example.

Response: In the revised manuscript, we have added hyperlink to each accession number in the section of Data Availability. Raw data of the long-read sequencing and short-read sequencing generated in this study have been deposited into the CNGBdb under accession code CNP0001815 (<https://db.cngb.org/search/project/CNP0001815/>). All 545 genome assemblies, 100 genome annotations (gff files), pan-genome, and VCF files (SNP, SV) have been also deposited in the CNGBdb with accession code CNP0002456 (<https://db.cngb.org/search/project/CNP0002456/>). This study also analyzed data for four previous released wild silkworm genomes that are available in the Sequence Read Archive (SRA) database according to accession numbers DRX054041 (<https://www.ncbi.nlm.nih.gov/sra/DRX054041>), DRX054040 (<https://www.ncbi.nlm.nih.gov/sra/DRX054040>), ERS402904 (<https://www.ncbi.nlm.nih.gov/sra/ERS402904>), ERS402902 (<https://www.ncbi.nlm.nih.gov/sra/ERS402902>).

Suggested changes, mainly grammatical:

I. 56: We suggest adding articles, such as here: "...the entire genomic content in THE

silkworm. We found that THE silkworm population..."

Response: We have corrected these grammatical errors in the revised manuscript. Please see line 56, "We construct a high-resolution pan-genome dataset representing almost the entire genomic content in the silkworm. We find that the silkworm population harbors a high density of genomic variants and identify 7,308 new genes, 4,260 (22%) core genes, and 3,432,266 non-redundant structure variations (SVs)."

I. 98: We suggest: "... depth of ~65x PER SAMPLE were..."

Response: We have added "per sample was" to this sentence. Please see Line 98, "A total of 31.52 Tb next-generation sequencing (NGS) reads with an average sequencing depth of ~65x per sample were obtained (Supplementary data 1)."

I. 113: "... represent a wide genetic diversity." Still not entirely clear what you mean.

Presumably you mean that these samples cover broadly across the diversity of domesticated silkworm and its wild progenitor species.

Response: Thank you for raising this point. We want to express that these samples cover broadly across the diversity of domesticated silkworm. In the revised manuscript, this sentence has been changed to "Of note, the genetic stock strains are widely distributed within the different subclades of the domestic silkworm clade (Fig. 2b) and cover therefore broadly across the diversity of domesticated silkworm". Please see Line 110-113.

I. 143-144: We think you should say something like "... were 98-99% on average, INCLUDING SINGLE COPY, DUPLICATED AS WELL AS FRAGMENTED GENES (Fig. 2f, ...". You do mention this in the methods, but it needs to be stated in the results.

Response: In the revised manuscript, we have added this information to this sentence. Please see Line 144-145, "The BUSCO evaluation value and mapping ratio of NGS reads to the assembled genomes were 98% and 99% on average, including single copy, duplicated as well as fragmented genes (Fig. 3f, Supplementary data 2), indicating that the assembled genomes have high completeness."

I. 233: "... the expression of gene..." -- either "we next investigated GENE EXPRESSION ..." or "EXPRESSION OF GENES ..."

Response: Thanks for your suggestion, this sentence has been changed to "We next investigated gene expression using RNA-seq data of 84 samples from fourteen strains that harbored 178,309 SVs in potential expression regulatory regions, forming 26,188 SV-gene pairs (for each pair, at least three strains with and three strains without the SV)". Please see Line 235.

II. 263-4: "Compared with A previous study..."

Response: We have added "a" to this sentence. Please see Line 263 of the revised manuscript. "Compared with a previous study (Xiang et al. 2018), 185 improvement-associated genes were newly identified."

I. 264: "... shared ONLY around 3% ..."

Response: We have added "ONLY" to this sentence in the revised manuscript. Please see line 265, "Interestingly, the two improved groups shared only around 3% of these improvement-associated regions (Fig. 5c)".

I. 266-7: "... the genetic bases of silkworm heterosis..." This seems unclear, unless these changes largely deleterious as homozygotes?

Response: In practice, hybrid silkworms produced by crossing CHN-I and JPN-I have substantial economic advantages. Despite the application of hybrid vigor in silkworm can be traced back to the early twentieth century, the genetic basis underpinning silkworm heterosis remains poorly understood. Here, we found that the two improved groups shared only around 3% of these improvement-associated regions (Fig. 5c), suggesting that breeding proceeded independently in CHN-I and JPN-I. We thus speculated that the genetic bases of heterosis between CHN-I and JPN-I could be partially improvement-associated gene complementation. In the revised manuscript, we have revised this sentence as "These results reveal parts of the genetic bases of silkworm heterosis and provide potential targets for improvement in silkworm breeding.". Please see line 267-269.

I. 395: " ... for survival. HOWEVER, only a few ..." It seems best to break a sentence here.

Response: Thanks for your suggestion. In the revised manuscript, the sentence has been changed to "Although the silkworm is a completely domesticated economic insect entirely dependent on humans for survival, only a few economically important genes are clearly identified so far.". Please see line 394-395.

I. 449: "... and without further SELECTIVE BREEDING..." -- "cultivation" seems wrong here, since it implies tilling earth and rearing plants rather than rearing insects.

Response: Thanks for your suggestion. "Cultivation" has been replaced by "selective breeding" in the revised manuscript. Please see line 447.

I. 452: "... further cultivated..." seems wrong here as well. Do you mean HIGHLY SELECTIVELY BRED strains?

Response: yes, "further cultivated" has been replaced by "highly selectively bred" in the revised manuscript. Please see line 453.

I. 456: "... represent THE ancestor (BOMBYX MANDARINA) of THE domesticated silkworm."

Response: In the revised manuscript, we have added "the" and "Bombyx mandarina" to this sentence. Please see line 457.

I. 574: "... (included complete ...". We think the authors mean "(INCLUDING complete ...". There are a number of other cases where "included" is used for "including" elsewhere that should also be corrected.

Response: In the revised manuscript, "included" has been replaced by "including". We

have carefully checked the full text and corrected other cases of similar grammatical errors in the revised manuscript.

Response to Reviewer #2

Reviewer #2 (Remarks to the Author):

I appreciate the authors' careful attention to the points I made in my initial review regarding organization and English expression and have only a few further substantive comments below, plus some additional minor suggestions for improving English usage which I noted on the attached annotated manuscript pdf rather than excerpting and describe them here. I especially appreciate the authors' change in reporting as "orthogroups" what they originally termed "multigene families." The former seems better suited to analysis and understanding of the relatively diverse and complex data they have developed for such a large and comprehensive pan-genome.

Response: Thank you very much for your comments and detailed revision on the attached annotated manuscript pdf. We have revised our manuscript based on your raised points in the manuscript pdf.

Without wanting to belabor their follow up analysis of "multigene families" too much (which I understand is being submitted privately by way of explanation and not for publication with the actual manuscript), although I appreciate the authors' re-worked data analysis of chorion genes shown in Response Figures 5, 6, and 7, and I'm not surprised by their finding that chorion protein gene numbers varied significantly among the strains they examined, still, the number of members in each of the major "families" they presented here for Dazao do not seem to be as well correlated as I would expect with the published data on chorion genes in this strain which was based on detailed manual annotation of BAC clones and analysis of expression patterns (e.g., please see Chen et al. (2015) SCIENTIFIC DATA | 2:150062 | DOI: 10.1038/sdata.2015.62). Perhaps this simply illustrates some pitfalls in using global bioinformatic or algorithmic tools rather than detailed manual annotation for accurate descriptions of these kinds of complex protein families which not only have overlapping functions but also a potential for relatively rapid evolution.

Response: We agree with your view. For identification of some complex protein families, the use of global bioinformatics or algorithmic tools may be less accurate than detailed manual annotation. However, for large-scale genomic data analysis, it is difficult to manually annotate each family in detail. This also implies that global bioinformatics tools and algorithms for genome-wide gene identification and annotation still require further improvement.